# Delta-dependent Notch activation closes the early neuroblast temporal program to promote lineage progression and neurogenesis termination in *Drosophila*

Chhavi Sood, Md Ausrafuggaman Nahid, Kendall R Branham, Matt Pahl, Susan E Doyle, Sarah E Siegrist*

Department of Biology, University of Virginia, Charlottesville, United States

*For correspondence: ses4gr@virginia.edu

Competing interest: The authors declare that no competing interests exist.

**Abstract** Neuroblasts in *Drosophila* divide asymmetrically, sequentially expressing a series of intrinsic factors to generate a diversity of neuron types. These intrinsic factors known as temporal factors dictate timing of neuroblast transitions in response to steroid hormone signaling and specify early versus late temporal fates in neuroblast neuron progeny. After completing their temporal programs, neuroblasts differentiate or die, finalizing both neuron number and type within each neuroblast lineage. From a screen aimed at identifying genes required to terminate neuroblast divisions, we identified Notch and Notch pathway components. When Notch is knocked down, neuroblasts maintain early temporal factor expression longer, delay late temporal factor expression, and continue dividing into adulthood. We find that Delta, expressed in cortex glia, neuroblasts, and after division, their GMC progeny, regulates neuroblast Notch activity. We also find that Delta in neuroblasts is expressed high early, low late, and is controlled by the intrinsic temporal program: early factor Imp promotes Delta, late factors Syp/E93 reduce Delta. Thus, in addition to systemic steroid hormone cues, forward lineage progression is controlled by local cell-cell signaling between neuroblasts and their cortex glia/GMC neighbors: Delta transactivates Notch in neuroblasts bringing the early temporal program and early temporal factor expression to a close.

## eLife assessment

This **useful** study reports on how Notch activity regulates the termination of neurogenesis in central brain during larval-pupal stages in *Drosophila*. The evidence supporting the claims is **solid**. The work will be of interest to developmental neurobiologists.

## Introduction

In most metazoans, termination of neurogenesis is an essential part of organism development, ensuring the formation of functional neural circuits and an adult brain of proper size and structure. Prolonged or ectopic neurogenesis can lead to cortical malformations and has been linked to neurodevelopmental disorders, including autism (*Hazlett et al., 2011*; *Marchetto et al., 2017*; *Patzlaff et al., 2018*; *Casingal et al., 2020*; *Ossola and Kalebic, 2022*). While the vast majority of neurons are generated during development, it still remains unclear how neurogenesis becomes progressively restricted and, in most cases, ends altogether after development is completed.

We use the genetically tractable model organism, *Drosophila melanogaster*, to determine how extrinsic cues, local and systemic, integrate with neural stem cell intrinsic cues to control neurogenesis timing and termination during development. The *Drosophila* CNS consists of two bilaterally symmetric

brain hemispheres and a ventral nerve cord that is functionally equivalent to the mammalian spinal cord. Each brain hemisphere contains an optic lobe and an equally sized central brain (CB) region that harbors distinctive neuropils for information processing. Neurons in the CB region are generated during development from the asymmetric cell divisions of a defined number of neural stem cells known as neuroblasts (NBs) in *Drosophila*. NBs in the CB region (referred to as CB NBs) are specified during embryogenesis and undergo stereotypic patterns of cell division (*Truman and Bate, 1988*; *Ito and Hotta, 1992*; *Doe, 2008*; *Siegrist et al., 2010*; *Homem and Knoblich, 2012*). Except for the mushroom body (MB) NB subset, all CB NBs enter and exit quiescence during the embryonic to larval transition and terminally differentiate or die 4–5 days later during early pupal stages (*Truman and Bate, 1988*; *Ito and Hotta, 1992*; *Doe, 2008*; *Maurange et al., 2008*; *Siegrist et al., 2010*; *Yang et al., 2017*).

Once CB NBs reactivate from quiescence in response to dietary nutrients, they divide continuously while changing gene expression over time (*Britton and Edgar, 1998*; *Chell and Brand, 2010*; *Sousa-Nunes et al., 2011*; *Liu et al., 2015*; *Syed et al., 2017*; *Yuan et al., 2020*). These controlled transitions of gene expression over time, referred to as temporal patterning, allow for a restricted set of neural stem cells to generate a pool of molecularly and functionally diverse neuron types (*Isshiki et al., 2001*; *Maurange et al., 2008*; *Bayraktar and Doe, 2013*; *Liu et al., 2015*; *Bahrampour et al., 2017*; *Ren et al., 2017*; *Syed et al., 2017*; *Miyares and Lee, 2019*). Early larval temporal factors include the Zinc finger transcription factor, Castor (Cas), the orphan nuclear receptor, Seven-up (Svp), the RNA-binding protein, IGF-II mRNA-binding protein (Imp), as well as others (*Maurange et al., 2008*; *Liu et al., 2015*; *Ren et al., 2017*; *Syed et al., 2017*). Svp expression primes NBs to respond to a systemic pulse of steroid hormone (ecdysone) during larval stages and switch temporal factor expression from early to late (*Ren et al., 2017*; *Syed et al., 2017*). Late temporal factors include the RNA-binding protein, Syncrip (Syp), the steroid hormone-induced transcription factor, Eip93F (E93), as well as others (*Liu et al., 2015*; *Syed et al., 2017*; *Pahl et al., 2019*). Imp (early) and Syp (late) mutually inhibit each other and are expressed in opposing gradients in NBs (*Liu et al., 2015*; *Yang et al., 2017*). Imp keeps CB NBs 'young' by inhibiting Syp and Mediator complex activity, whereas Syp inhibits Imp and promotes nuclear accumulation of the pro-differentiation transcription factor Prospero (Pros) in most CB NBs (*Homem et al., 2014*; *Liu et al., 2015*; *Yang et al., 2017*). During early pupal stages, CB NBs undergo reductive divisions and terminally differentiate, except for the MB NB subset, which divides several days longer and undergo autophagy/apoptosis prior to adult eclosion (*Maurange et al., 2008*; *Siegrist et al., 2010*; *Homem et al., 2014*; *Pahl et al., 2019*). Independent of neurogenesis timing and the mechanism by which CB NB stop divisions, temporal patterning plays a key role in controlling numbers and types of neurons made within each of the NB lineages (*Maurange et al., 2008*; *Tsuji et al., 2008*; *Bahrampour et al., 2017*; *Yang et al., 2017*; *Pahl et al., 2019*).

From a targeted RNAi screen aimed at identifying genes required to terminate CB NB divisions and neurogenesis, we identified Notch and Notch pathway components. Notch is an evolutionarily conserved cell-cell signaling pathway classically known for regulating binary cell fate decisions, 'A' versus 'B' (*Muskavitch, 1994*; *Cau and Blader, 2009*). Here, we show that Notch signaling also regulates binary temporal decisions, 'early' versus 'late'. In *Drosophila*, there is one Notch receptor and two ligands, Delta (Dl) and Serrate (Ser). Notch receptor is proteolytically cleaved after ligand binding, first by Kuzbanian (Kuz), an ADAM metalloprotease, and then by γ-secretase. Cleaved Notch ICD (intracellular domain) relocates to the nucleus where it binds to Suppressor of Hairless [Su(H)] and Mastermind to regulate gene expression (*Rebay et al., 1991*; *Fortini and Artavanis-Tsakonas, 1994*; *Pan and Rubin, 1997*; *De Strooper et al., 1999*; *Mumm et al., 2000*; *Kitagawa et al., 2001*; *Kopan and Ilagan, 2009*). We recently reported that Notch signaling regulates CB NB quiescence during the embryonic to larval transition (*Sood et al., 2022*). When Notch is knocked down, some CB NBs continue dividing during this transition. We also reported that Notch activity becomes attenuated in quiescent CB NBs because CB NBs are no longer dividing and producing Delta-expressing GMC daughters for Notch pathway transactivation. Moreover, low Notch is necessary for CB NBs to reactivate from quiescence in response to dietary nutrients (*Sood et al., 2022*).

Here, we report that Notch signaling also regulates neurogenesis termination during pupal stages. When Notch is knocked down, CB NBs maintain early temporal factor expression longer resulting in a delay of late temporal factor expression with prolonged neurogenesis into late pupal stages and

early adulthood. This defect in temporal patterning (switching from early to late) occurs well after CB NB exit from quiescence suggesting that Notch is required at multiple times throughout development in controlling CB NB proliferation decisions. Furthermore, we determine that Delta is the Notch ligand that activates Notch in CB NBs and reductions in Delta also lead to defects in CB NB temporal patterning. Moreover, we find that Delta in CB NBs, which is segregated to GMCs after cell division to transactivate Notch, is regulated by CB NB temporal factors. Early factor Imp promotes Delta, whereas late factors Syp and E93 reduce Delta. Together, we report that Notch signaling positively regulates forward lineage progression by closing off the early temporal window and control of Notch pathway activity is regulated by CB NB intrinsic temporal factors.

## Results

### Notch signaling is required for CB NB elimination and termination of neurogenesis

All CB NBs, except the MB NB subset (four per brain hemisphere) terminally differentiate or die during early pupal stages (*Figure 1A*; *Truman and Bate, 1988*; *Ito and Hotta, 1992*; *Maurange et al., 2008*; *Siegrist et al., 2010*; *Homem et al., 2014*; *Yang et al., 2017*). MB NBs divide several days longer and undergo apoptotic/autophagic cell death shortly before adult eclosion (*Figure 1A and B*; *Siegrist et al., 2010*; *Pahl et al., 2019*). No CB NBs remain in adult animals and no new neurons are produced (*Figure 1A and B*; *Truman and Bate, 1988*; *Siegrist et al., 2010*; *Yang et al., 2017*). From a targeted RNAi screen aimed at identifying genes required to terminate NB divisions and neurogenesis, we identified Notch (N) and Notch pathway components. At 48 hr APF (after pupal formation), midway through pupal stages, control animals have only the four MB NBs remaining in each brain hemisphere (*Figure 1A and B*). In contrast when Notch was knocked down in NBs (*worGAL4,UAS-N RNAi #HMS00001*), on average five additional CB NBs remained (*Figure 1C and D*). In 1-day old *N RNAi* adults, CB NBs were also present, but not the MB NBs (*Figure 1C and D*). Ectopically persisting *N RNAi* CB NBs (CB NBs at 48 hr APF and beyond) expressed the NB transcription factor Deadpan (Dpn), the S-phase indicator *pcnaGFP*, and were small on average compared to control CB NBs during earlier developmental stages (L3 control, average diameter 10–15 μm) (*Figure 1B, C, and E*). However, at 30 hr APF when control CB NBs are still present, *N RNAi* CB NBs were larger on average (*Figure 1B, C, and E*). To confirm the *N RNAi* phenotype, we used MARCM to generate CB NB clones mutant for the Notch loss-of-function allele, *Notch[55e11]* (*Lee and Luo, 1999*; *Lehmann et al., 1983*). Animals were heat shocked at freshly hatched larval stages and brains assayed at 48 hr APF. More than 50% of the GFP positive clones had a single Dpn positive NB (*Figure 1F*). In contrast, control clones had no Dpn positive NBs. We conclude that Notch functions in a lineage-dependent manner to eliminate CB NBs and terminate neurogenesis.

Next, we assayed other Notch pathway components. Following knockdown of *kuz* (*worGAL4,UAS-kuzRNAi #HMS05424*), CB NBs, other than the MB NBs, remained at 48 hr APF and in adults (*Figure 1G, I, and J*). Following *Su(H)* knockdown (*worGAL4,UAS-Su(H)RNAi #HMS05748*), CB NBs, other than the MB NBs, also remained (*Figure 1H and J*). Similar to the *N RNAi* phenotype, ectopically persisting CB NBs expressed Dpn and *pcnaGFP*, and were small (*Figure 1K*). We conclude that the evolutionarily conserved Notch cell signaling pathway is required for CB NB elimination and neurogenesis termination.

### Delta expressed in neighboring GMCs and cortex glia regulates Notch activity in CB NBs

Next, we assayed the expression of Delta (Dl) and Serrate (Ser), two Notch ligands that activate Notch signaling when expressed on neighboring cells. Using a *Delta-GFP* protein trap line, we found that Delta was expressed in CB NBs and their recently born Prospero (Pros) positive GMC progeny during larval stages, consistent with previous reports (*Figure 2A and B*; *Kooh et al., 1993*; *Sood et al., 2022*). Delta was also expressed in cortex glia, a glial subset that ensheathe CB NBs and their GMC progeny, but levels were relatively low (*Figure 2C*; *Hayashi et al., 2002*; *Yuan et al., 2020*). Using a *Serrate-GFP* protein trap line, we found that Serrate was expressed in cortex glia, but not in CB NBs nor their GMC progeny (*Figure 2D–F*). Next, we knocked down each of the Notch ligands to determine which Notch ligand from what cell type regulates Notch activity in CB NBs. We used

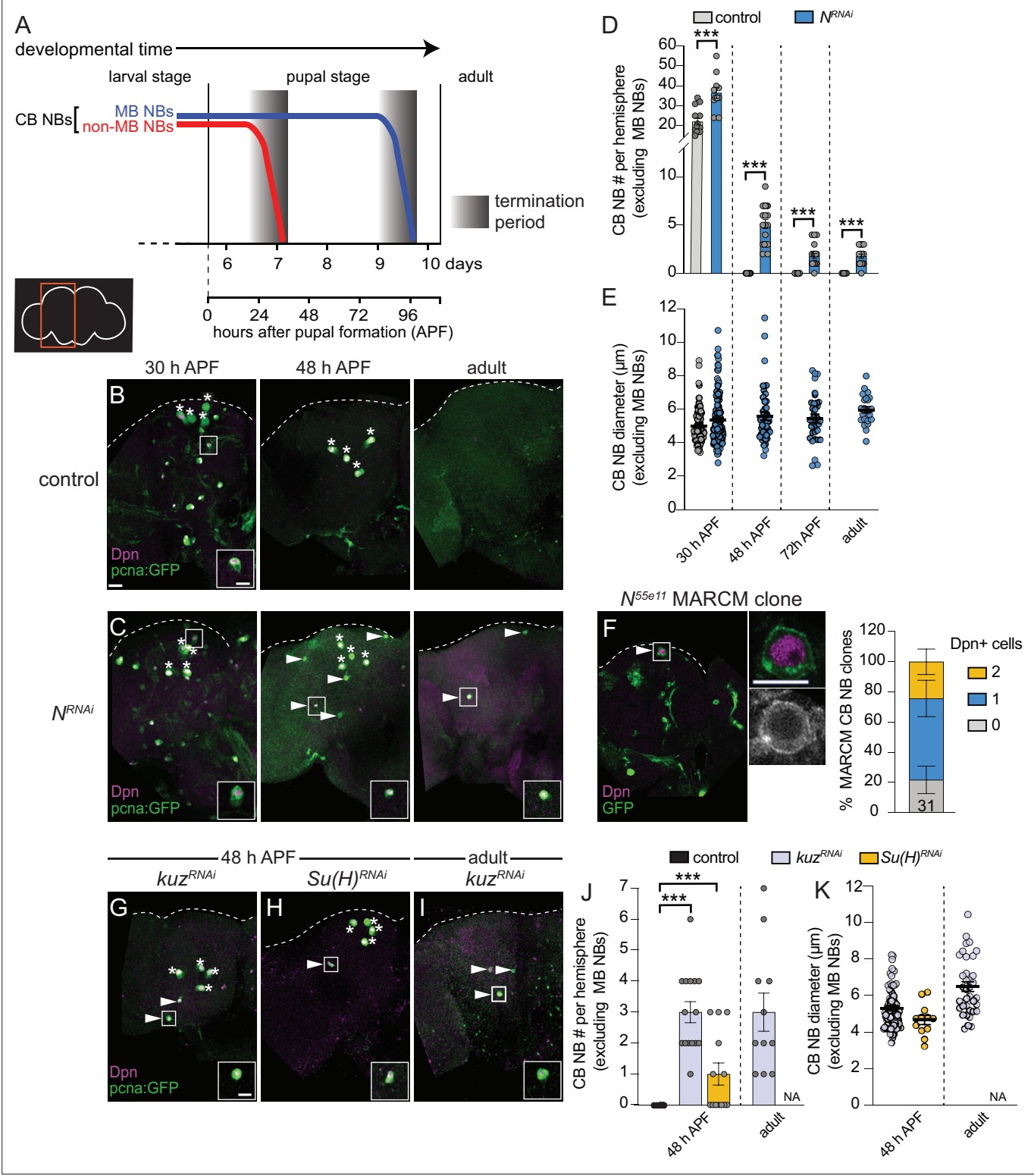

**Figure 1.** Notch signaling regulates timing of central brain neuroblast (CB NB) elimination and neurogenesis termination. (**A**) Schematic showing developmental timeframe of CB NB elimination with timeline below used for developmental staging. (**B–C, G–I**) Maximum intensity projections of single brain hemispheres from indicated genotypes, times, with markers listed in bottom left. Asterisks indicate the four mushroom body (MB) NBs and arrowheads indicate some of the ectopically proliferating CB NBs (non-MB NBs). One ectopic CB NB (white box) shown at higher magnification in bottom right. (**D, J**) Quantification of CB NB number (excluding MB NBs) per brain hemisphere at indicated times and genotypes. Each data point

*Figure 1 continued on next page*

*Figure 1 continued*

represents one brain hemisphere, mean ± SEM, ***p-value ≤0.001 (unpaired two-tailed Student's t-test). (**E, K**) Quantification of average CB NB diameter, used as a proxy for NB size, at indicated times and genotypes. Each data point equals one CB NB (n≥4 animals per genotype), mean ± SEM. (**F**) Single optical section of a brain hemisphere from indicated genotype at 48 hr APF (after pupal formation) with markers listed in panels with high magnification panel to right of ectopic CB NB in white box. Distribution of *Notch*^55e11 MARCM CB NB clones containing Dpn positive NBs. Scale bar equals 20 μm (panels) or 10 μm (insets) in this and all subsequent figures. Panel genotypes listed in **Supplementary file 1**.

the NB-specific *E(spl)mγ-GFP* reporter to assay Notch activity (**Furriols and Bray, 2001**; **Almeida and Bray, 2005**; **Zacharioudaki et al., 2012**). In controls, CB NBs express *E(spl)mγ-GFP* and following knockdown of Notch, *E(spl)mγ-GFP* was not expressed (**Figure 2G and H**). Next, we used *worGAL4* to knock down Delta in CB NBs and because GAL4 is inherited after CB NBs divide, neighboring GMC progeny as well. Following knockdown of Delta (*worGAL4,UAS-Dl RNAi #HMS01309*), *E(spl)mγ-GFP*

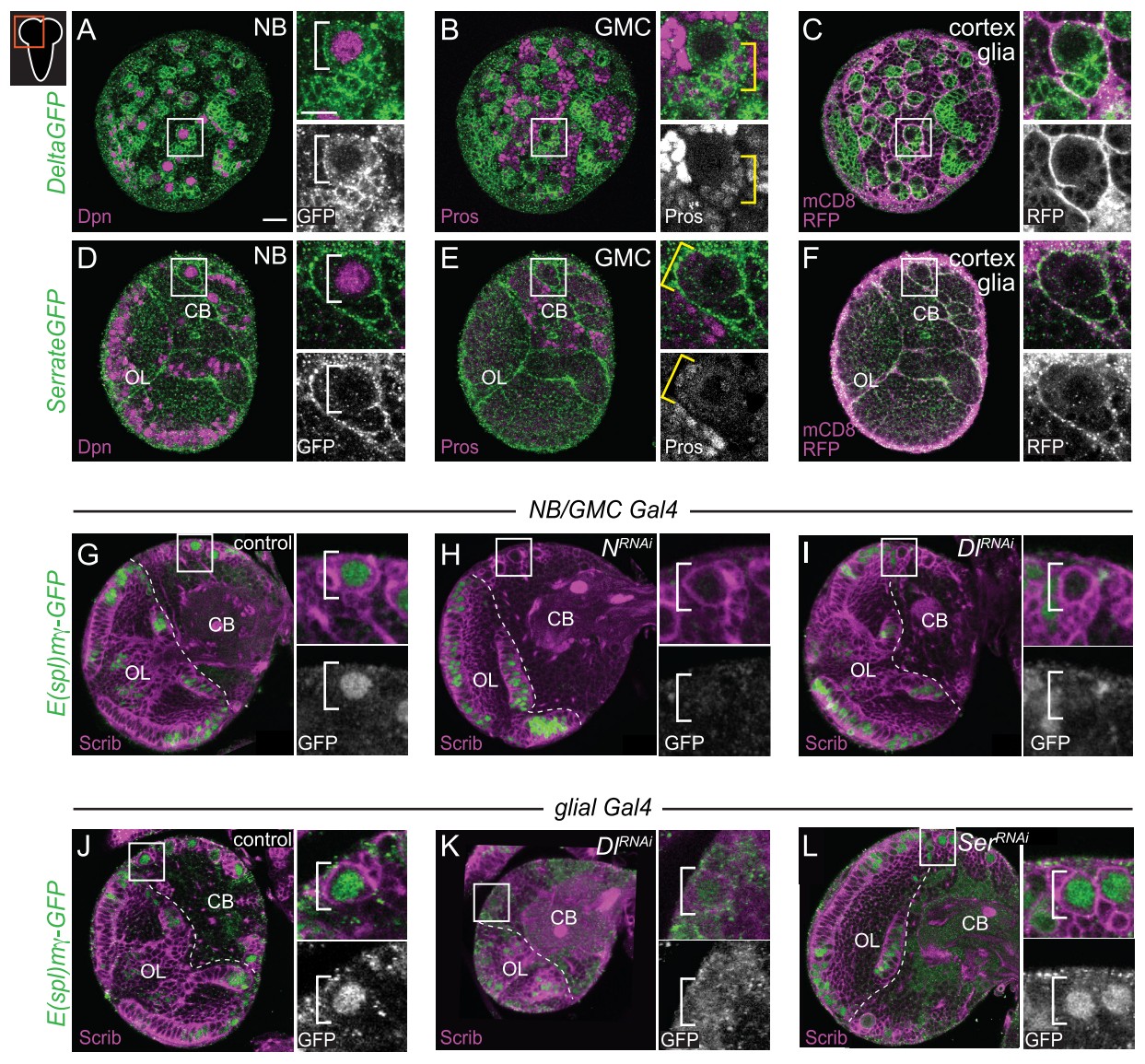

**Figure 2.** Delta expressed in CB neuroblasts (NBs), GMCs, and cortex glia regulates CB NB Notch activity. (**A–F**) Single optical section of a brain hemisphere from the indicated genotypes at wandering L3 stages. Higher magnification image of the CB NB highlighted by the white box is shown to the right of the colored overlays. Top panels are higher magnification colored overlay with single channel grayscale images below. White brackets indicate the CB NB and yellow brackets indicate newborn GMC progeny. (**G–L**) Single optical section of a brain hemisphere from the indicated genotypes at 72 hr ALH. Higher magnification image of the CB NB highlighted by the white box is shown to the right of the colored overlays. Scale bar equals 20 μm (panels) and 10 μm (insets). CB: central brain; OL: optic lobe. Panel genotypes listed in **Supplementary file 1**.

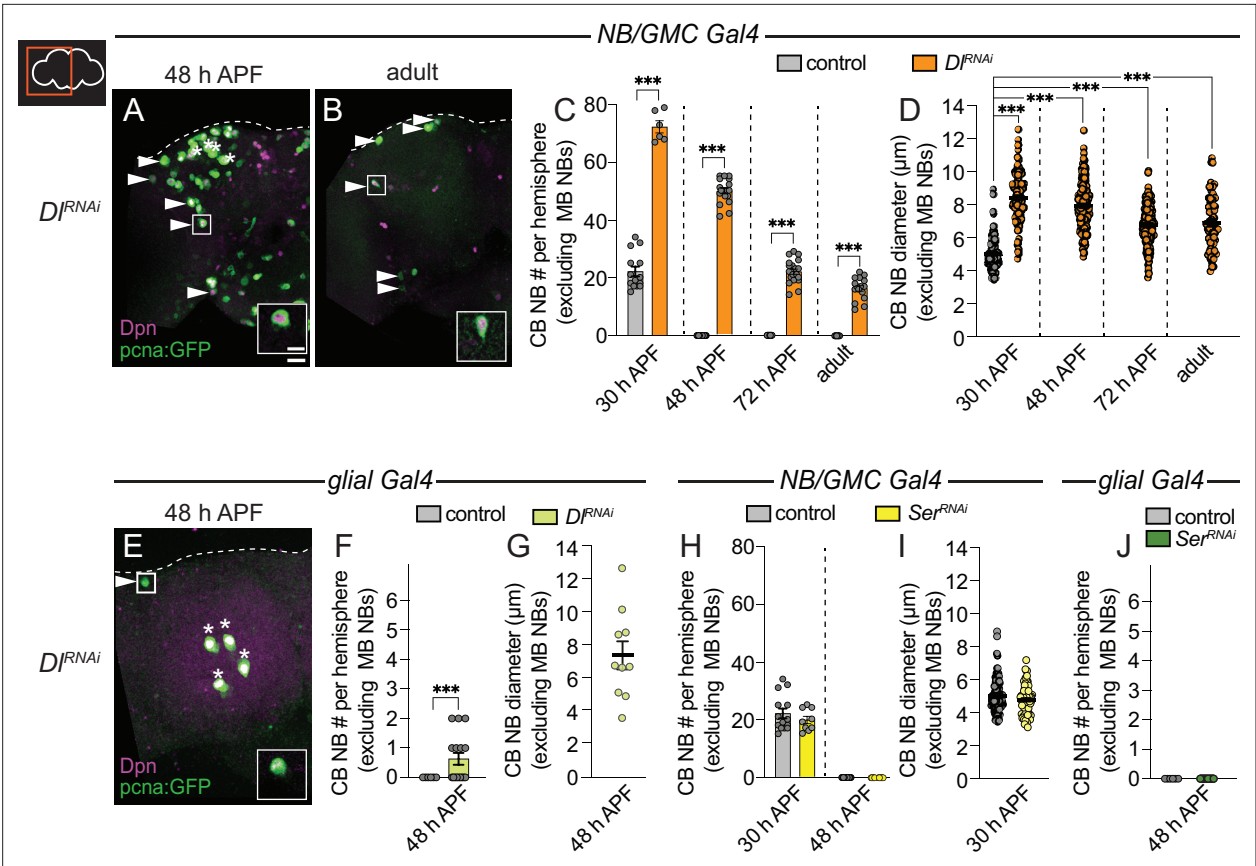

**Figure 3.** Delta is required to eliminate central brain neuroblasts (CB NBs) and terminate neurogenesis. (**A, B, E**) Maximum intensity projections of single brain hemispheres from indicated genotypes. Asterisks indicate the mushroom body (MB) NBs and the white arrowheads indicate some of the ectopically proliferating CB NBs. Inset shows a higher magnification of the ectopically proliferating CB NB highlighted by the white box. (**C, F, H, J**) Quantification of CB NB number (excluding the MB NBs). Each data point represents one brain hemisphere. Control data in (**C**) is the same as *Figure 1D*. Mean ± SEM. ***p≤0.001 (unpaired two-tailed Student's t-test). (**D, G, I**) Quantification of CB NB size (excluding the MB NBs) in the indicated genotypes and developmental times. Each data point represents one NB (n≥4 animals per genotype). Control data in (**D**) is the same as *Figure 1E*. Mean ± SEM. ***p≤0.001, *p≤0.033 (Kruskal-Wallis test). Scale bar equals 20 µm (panels) and 10 µm (insets). Panel genotypes listed in *Supplementary file 1*.

was not detected in CB NBs (*Figure 2I*). Next, we used a pan-glial GAL4 line to knock down either Delta or Serrate in cortex glia. Following knockdown of Delta (*repoGAL4,UAS-Dl RNAi #HMS01309*), *E(spl)mγ-GFP* was reduced compared to controls (*Figure 2J and K*). In contrast, following Serrate knockdown (*repoGAL4,UAS-SerRNAi #HMS01179*), *E(spl)mγ-GFP* was not affected (*Figure 2L*). We conclude that Delta is the primary Notch ligand expressed in CB NBs and their GMC progeny, while both Delta and Serrate are expressed in neighboring cortex glia. Moreover, Delta regulates CB NB Notch activity.

## Delta-dependent Notch activation is required for CB NB elimination and termination of neurogenesis

Next, we knocked down each of the Notch ligands in neighboring cell types and assayed CB NB number during pupal stages. When Delta was knocked down in NBs and GMC progeny (*worGAL4,UAS-Dl RNAi #HMS01309*), ectopically persisting CB NBs were found in brains at all stages examined and in young adults (*Figure 3A–C*). When Delta was knocked down in cortex glia (*NP0577GAL4,UAS-Dl RNAi #HMS01309*), one, occasionally two ectopically persisting CB NBs were found at mid pupal stages (*Figure 3E and F*). Moreover, CB NBs that ectopically persisted tended to be larger than control CB NBs at early stages, consistent with the notion that cell size correlates with timing of termination (*Figure 3D and G*; *Maurange et al., 2008*; *Siegrist et al., 2010*; *Homem et al., 2014*; *Yang et al.,*

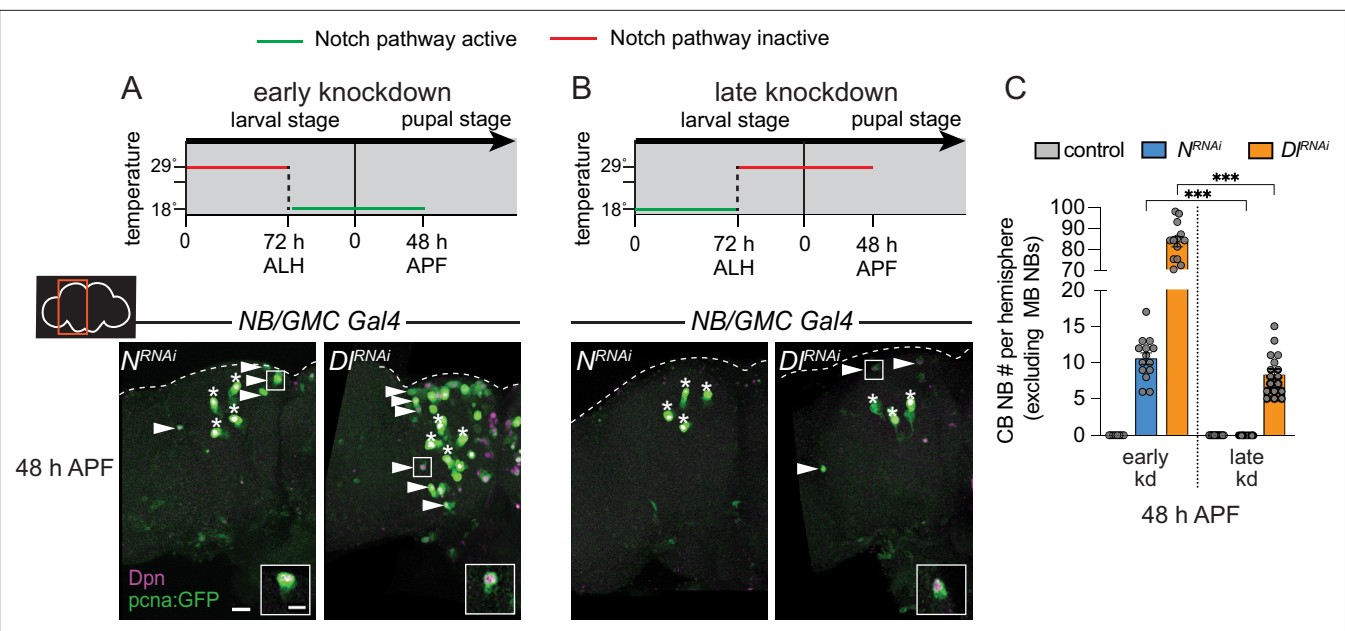

**Figure 4.** Notch is required early to eliminate central brain neuroblasts (CB NBs). (**A, B**) Top, schematic depicting experimental setup to temporally control knockdown of Notch signaling. Bottom, maximum intensity projections of single brain hemispheres from indicated genotypes at 48 hr APF (after pupal formation) relative to 25°C. Asterisks indicate the mushroom body (MB) NBs and the white arrowheads indicate some of the ectopically proliferating CB NBs. Inset shows a higher magnification of an ectopically proliferating CB NB highlighted by the white box. (**C**) Quantification of CB NB number (excluding the MB NBs) in the indicated genotypes. Each data point represents one brain hemisphere. Mean ± SEM. ***p≤0.001 (Welch's t-test). Scale bar equals 20 μm (panels) and 10 μm (insets). Panel genotypes listed in ***Supplementary file 1***.

The online version of this article includes the following figure supplement(s) for figure 4:

**Figure supplement 1.** Temporal control of Notch activity in central brain neuroblasts (CB NBs).

*2017*). Next, although Serrate was not detected in CB NBs nor their GMC progeny, we still assayed CB NB number and size after Serrate knock down in both NBs and their GMC progeny (*worGAL4,UAS-SerRNAi #HMS01179*). No differences were observed compared to controls (***Figure 3H, I***). Next, we knocked down Serrate in glia (*repoGAL4,UAS-SerRNAi #HMS01179*) and again no differences were found compared to controls (***Figure 3J***). We conclude that Delta, but not Serrate, is required for CB NB elimination and termination of neurogenesis.

## Early Notch signaling is required to terminate neurogenesis during pupal stages

Next, we used a temperature-sensitive GAL80 to determine when during development Notch signaling is required to eliminate CB NBs and terminate neurogenesis. Animals were raised at 29°C (GAL80 inactive, GAL4 active) until 72 hr ALH and then switched to 18°C (GAL80 active, GAL4 inactive) or the converse (***Figure 4A and B***). In control animals at 48 hr APF, after either temperature shift regime, only the four MB NBs were present in each brain hemisphere (***Figure 4C*** and data not shown). When Notch or Delta knockdown animals were raised at 29°C (Notch pathway inactive) and then switched to 18°C late (Notch pathway active), a significant number of persisting CB NBs were found at 48 hr APF (***Figure 4A and C***). In contrast, when animals were raised at 18°C (Notch pathway active) and then switched to 29°C late (Notch pathway inactive), no or significantly fewer persisting CB NBs were found (***Figure 4B and C***). Absence or presence of Notch pathway activity under each temperature shift regime was verified using *E(spl)mγ-GFP* reporter expression (***Figure 4—figure supplement 1A and B***). We conclude that early Notch pathway activity is required to eliminate CB NBs and terminate neurogenesis.

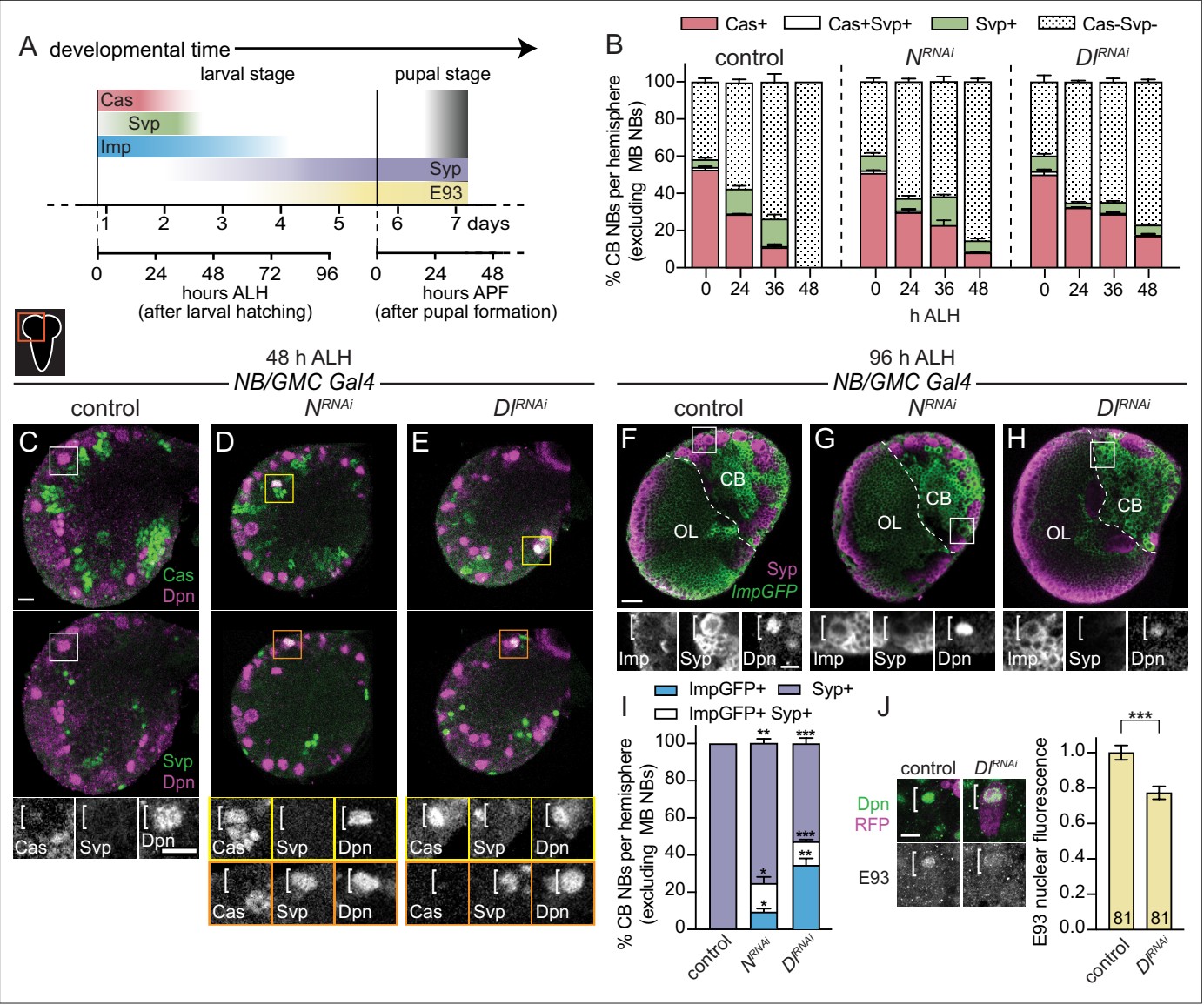

**Figure 5.** Notch signaling refines temporal factor expression boundaries. (**A**) Schematic of temporal factor expression in CB neuroblasts (NBs) during larval and pupal development. Top timeline (days) refers to developmental timing with two timelines below used for developmental staging. Larva hatch 22 hr after egg lay. (**B, I**) Quantification of the percentage of CB NBs (excluding the mushroom body [MB] NBs) in the indicated genotypes and developmental times expressing the indicated temporal factors. Mean ± SEM. n≥3 animals, ***p≤0.001, **p≤0.002, *p≤0.033 (two-way ANOVA). (**C–H**) Single optical section of a brain hemisphere from the indicated genotypes and developmental times. Higher magnification image of the CB NB highlighted by the white box is shown below the colored overlays. White brackets indicate the CB NB. (**J**) Single optical section of a CB NB, colored overlay with grayscale image below. White brackets indicate the CB NB with quantification of normalized nuclear E93 intensities. Column numbers indicate the number of CB NB clones (excluding the MB NBs) scored. Mean ± SEM. ***p≤0.001 (Mann-Whitney test). Scale bar equals 20 µm (panels) and 10 µm in single CB NB panels. CB: central brain; OL: optic lobe. Panel genotypes listed in *Supplementary file 1*.

The online version of this article includes the following figure supplement(s) for figure 5:

**Figure supplement 1.** Imp expression is prolonged and Syp expression delayed in animals with reduced Notch pathway activity.

## Delta-dependent Notch activation refines temporal boundaries by closing the early CB NB temporal window

During development, CB NBs sequentially express a series of intrinsic factors over time to generate a diversity of neuron types (*Figure 5A*). Defects in intrinsic temporal factor expression lead to changes in the molecular composition of neuron types produced and defects in timing of CB NB elimination and neurogenesis termination (*Isshiki et al., 2001*; *Maurange et al., 2008*; *Liu et al., 2015*; *Ren*

*et al., 2017*; *Syed et al., 2017*; *Yang et al., 2017*; *Pahl et al., 2019*). Because Notch signaling is required early to eliminate CB NBs, we assayed expression of early temporal factors, Cas and Svp. In freshy hatched control larvae (0 hr ALH), approximately 50% of CB NBs expressed Cas and 5% expressed Svp (*Figure 5B*). Over time, the percentage of Cas expressing CB NBs declined, while Svp expressing CB NBs modestly increased (*Figure 5B*). Less than 1% of CB NBs co-expressed Cas and Svp at any stage and expression of both factors was absent by 48 hr ALH (*Figure 5B and C*). This is consistent with work published previously (*Isshiki et al., 2001*; *Tsuji et al., 2008*; *Chai et al., 2013*; *Maurange et al., 2008*; *Ren et al., 2017*; *Syed et al., 2017*). When Notch or Delta were knocked down in CB NBs and their GMC progeny, approximately 50% of CB NBs expressed Cas at freshly hatched larval stages (0 hr ALH), same as controls, and slightly more expressed Svp (*Figure 5B*). Over time, the percentage of Cas expressing CB NBs declined, while Svp expressing CB NBs remained relatively unchanged or reduced (*Figure 5B*). At 48 hr ALH, in contrast to controls, Cas and Svp were still expressed in CB NBs in both Notch and Delta knockdown animals (*Figure 5B, D, and E*). This suggests that Notch signaling is required for the cessation of early temporal factor expression. Moreover, early temporal defects could lead to defects in later temporal factor expression.

Next, we looked at later developmental time points. In control animals at 72 hr ALH, some CB NBs still expressed Imp, but most had transitioned to expressing Syp in response to steroid hormone signaling, and by 96 hr ALH, all expressed Syp and E93 (except MB NBs) (*Figure 5F, I, and J* and *Figure 5—figure supplement 1A and D*). This is consistent with work published previously (*Liu et al., 2015*; *Ren et al., 2017*; *Syed et al., 2017*; *Yang et al., 2017*). When Notch or Delta were knocked down in NBs and their GMC progeny, we found that CB NBs still expressed Imp at both 72 and 96 hr ALH (*Figure 5G–I* and *Figure 5—figure supplement 1B–D*). Coincident with prolonged Imp expression, we found a reduction in CB NBs expressing Syp and many co-expressed both Imp and Syp, a phenotype not seen in control animals (*Figure 5G–I* and *Figure 5—figure supplement 1B–D*). Next, we assayed E93, whose expression is dependent on ecdysone signaling and in the MB NB lineage, Syp (*Syed et al., 2017*; *Pahl et al., 2019*). We generated RFP expressing CB NB clones that co-express *Dl RNAi* and found that E93 protein levels were reduced more than 20% compared to controls (*Figure 5J*). We conclude that Delta-dependent Notch activation is required to sharpen the boundaries of temporal factor expression. Moreover, these defects in temporal factor boundaries could lead to defects in CB NB elimination.

Defects in timing of temporal transitions could be due to defects in cell cycle progression, although embryonic NBs still transition independent of cell division (*Grosskortenhaus et al., 2005*). We used PH3 to assay CB NB mitotic activity. In Delta knockdown animals, the percentage of PH3 positive CB NBs was reduced compared to control (*Figure 5—figure supplement 1E*). At 48 hr APF however, Delta knockdown CB NBs were still dividing based on PH3 expression (*Figure 5—figure supplement 1F*). To determine whether CB NBs ectopically persist due to defects in cell cycle rate, we co-expressed *dp110* to constitutively activate PI3-kinase in Delta knockdown animals. A significant number of *pcnaGFP* expressing, Dpn positive CB NBs were still observed, suggesting that defects in cell cycle timing and growth rates alone cannot account for ectopic persistence of CB NBs into later developmental stages and adulthood (*Figure 5—figure supplement 1G*).

## CB NBs with reduced Notch pathway activity persist into adulthood due to temporal patterning defects

Next, we assayed temporal factor expression in CB NBs that ectopically persisted into late pupal stages. We generated *Notch*[55e11] MARCM CB NB clones as described previously and found that ectopically persisting CB NBs expressed either Imp alone, co-expressed both Imp and Syp, or expressed Syp alone similar to CB NBs at earlier larval stages (*Figure 6A*, n=14 clones). Similar expression profiles were observed in ectopically persisting CB NBs in Delta knockdown animals (*worGAL4, UAS-Dl RNAi #HMS01309*) (*Figure 6B*). Next, we tested whether temporal patterning defects account for the ectopic persistence of CB NBs with reduced Notch pathway activity. First, we knocked down Imp. Knocking down Imp alone leads to premature CB NB loss due to premature expression of late temporal factors (*Figure 6—figure supplement 1A and B*; *Yang et al., 2017*). When Imp was knocked down together with Delta (*worGAL4, UAS-Dl RNAi #HMS01309, UAS-Imp RNAi#HMS01168*), CB NB number was significantly reduced compared to Delta knockdown alone (*Figure 6C, D, and G*). CB NB size was also reduced, consistent with previous work demonstrating the importance of the early

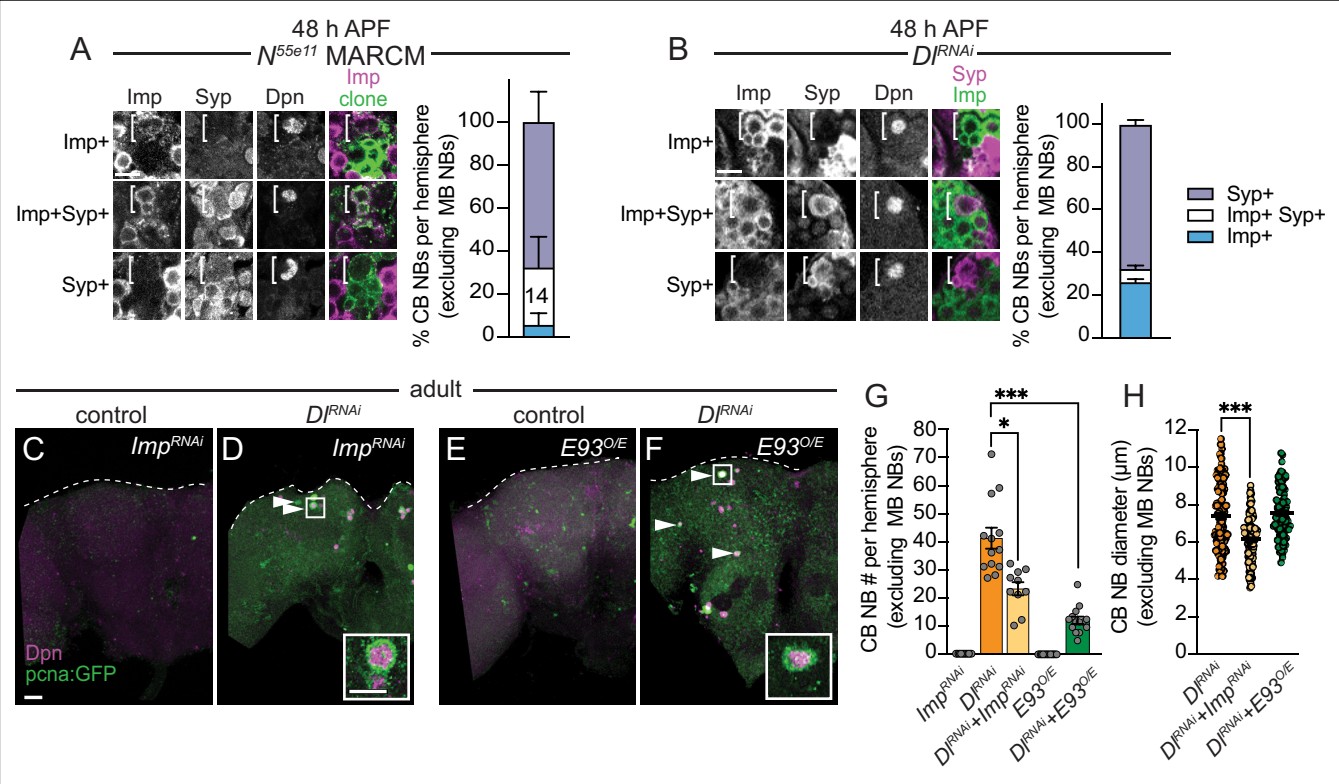

**Figure 6.** Central brain neuroblasts (CB NBs) ectopically persist due to prolonged early factor Imp expression and reduced late factor E93 expression. (**A, B**) Single optical sections of CB NBs from indicated genotypes and developmental times. Single channel grayscale images with colored overlay to the right. White brackets indicate CB NBs. Right, percentage of CB NBs (excluding the mushroom body [MB] NBs) in the indicated genotypes and developmental times expressing the indicated temporal factors. Column number (**A**) indicates number of clones scored. Mean ± SEM. n≥3 animals (**B**). (**C–F**) Maximum intensity projections of single brain hemispheres from indicated genotypes in 1-day old adults. White arrowheads indicate some ectopically proliferating CB NBs (MB NBs are absent). Inset shows a higher magnification of an ectopically proliferating CB NB highlighted by the white box. (**G**) Quantification of CB NB number (excluding MB NBs) in the indicated genotypes. Each data point represents one brain hemisphere. Mean ± SEM. ***p≤0.001, *p≤0.033 (Kruskal-Wallis ANOVA). (**H**) Quantification of CB NB size (excluding the MB NBs) in the indicated genotypes. Each data point represents one CB NB (n≥4 animals per genotype). Mean ± SEM. ***p≤0.001, *p≤0.033 (Kruskal-Wallis ANOVA). Scale bar equals 20 µm (panels) and 10 µm (insets). Panel genotypes listed in ***Supplementary file 1***.

The online version of this article includes the following figure supplement(s) for figure 6:

**Figure supplement 1.** Defects in temporal patterning account for defects in timing of central brain neuroblast (CB NB) elimination and neurogenesis termination when Notch activity is reduced.

factor Imp in promoting growth (***Figure 6H***; ***Yang et al., 2017***). Next, we constitutively expressed the late temporal factor E93, since E93 levels were reduced in *Dl RNAi* CB NB clones. When E93 was constitutively expressed in Delta knockdown animals, a significant reduction in CB NB number was observed (***Figure 6E–H***). We conclude that ectopically persisting CB NBs in animals with reduced Notch pathway activity is due to defects in CB NB temporal patterning: Imp expression is prolonged and E93 levels are reduced.

## The early temporal factor Imp positively regulates Delta expression

To better understand how Notch signaling controls CB NB temporal factor expression, we mined publicly available datasets. The datasets that we mined include (1) results from Notch genetic, molecular, and biochemical interaction studies (https://flybase.org/reports/FBgn0004647), (2) results from RIP-seq (RNA immunoprecipitation) experiments using Imp or Syp as bait (***McDermott et al., 2014***; ***Samuels et al., 2020***), and (3) RNA-sequence data from isolated, pooled AL (antennal lobe) NBs at early and late timepoints (***Liu et al., 2015***). First, we identified genes that were differentially expressed (DEGs) in AL NBs early (24 hr ALH) versus late (84 hr ALH). We identified 1861 genes (adj. p-value <0.1 and log fold change >1), including known temporal factors (***Figure 7A***). Next, we asked which if

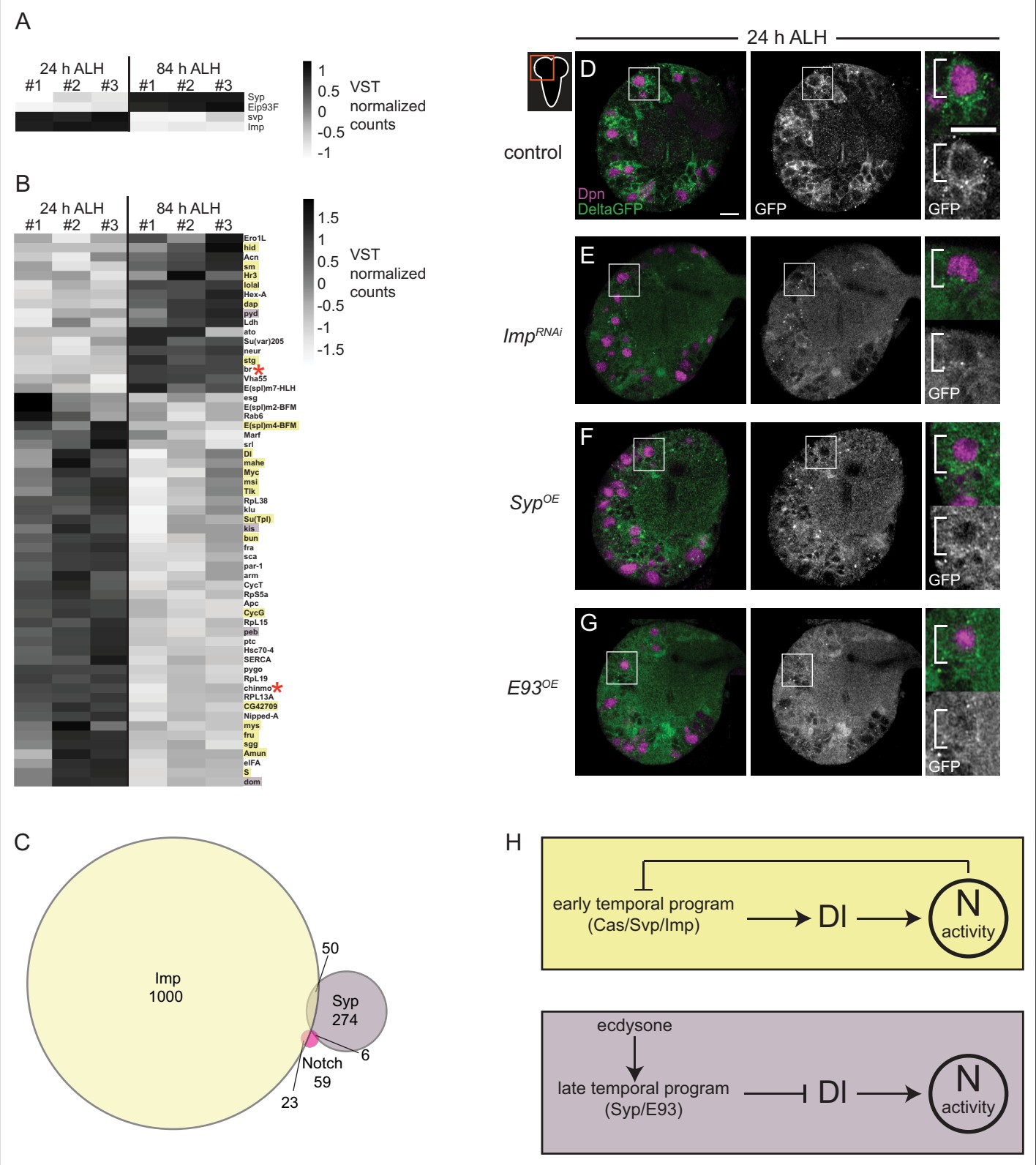

**Figure 7.** Delta is expressed at higher levels early and is positively regulated by the early temporal factor Imp. (**A–B**) Heatmap showing the list of genes that are differentially expressed in the AL neuroblasts (NBs) from 24 to 84 hr ALH. In (**B**) genes are color coded to show if they are also targets of either Imp (yellow) or Syp (purple) or both (red asterisk). Variance-stabilizing transformation (VST) normalized counts were used to plot the heatmaps. (**C**) Venn diagram showing the number of target genes analyzed that are common between Notch, Imp, and Syp. (**D–G**) Single optical section of a brain

*Figure 7 continued on next page*

*Figure 7 continued*

hemisphere from the indicated genotypes at 24 hr ALH expressing *Delta-GFP*. Higher magnification image of the central brain (CB) NB highlighted by the white box is shown to the right of the single channel grayscale images. Top panels are higher magnification colored overlay with single channel grayscale images below. White brackets indicate the NB. Scale bar equals 10 µm. Panel genotypes listed in ***Supplementary file 1***. (**H**) Model of Delta-dependent Notch activation in regulation of CB NB temporal patterning.

any were present in datasets from Notch genetic, molecular, and biochemical interaction studies. We identified 59 genes (***Figure 7B and C***). Next, we determined that 23 of these 59 genes were found in the Imp-RIP dataset (***Figure 7B***, orange highlight) and 6 in the Syp-RIP dataset (***Figure 7B***, purple highlight). To our surprise, Delta was one of the genes on this gene list. Delta transcript levels were expressed high early, and Delta mRNA co-immunoprecipitated with Imp in RIP-seq experiments. This raised the possibility that Delta, a known Notch target gene, is regulated by NB temporal factors (***Zhang et al., 2021***). To test this possibility, we knocked down Imp in CB NBs (*worGAL4,UAS-Imp-RNAi #HMS01168*) and assayed expression of the protein trap, *Delta-GFP*. At 24 hr ALH, *Delta-GFP* levels were reduced compared to controls (***Figure 7D and E***). This suggests that Imp positively regulates Delta.

Next, we asked whether late temporal factors would also regulate Delta, since Delta transcript levels decrease over time. We assayed *Delta-GFP* in CB NBs following constitutive Syp or E93 expression. We examined brains at 24 hr ALH, a time when Syp and E93 are normally not expressed and found reduced *Delta-GFP* expression in NBs and their GMC progeny compared to controls under both conditions (***Figure 7F and G***). Together with the reduction in Delta transcript levels during late larval stages (***Figure 7B***), these results suggest that Delta and Delta-dependent Notch transactivation are regulated by CB NB intrinsic temporal factors.

## Discussion

Here, we report that CB NBs utilize Notch signaling to progress forward through their stem cell lineages, ultimately terminating their divisions through differentiation or death (see Model ***Figure 7H***). Somewhat paradoxically, we find that early Notch activity is required early to terminate CB NB divisions late. This is because Notch regulates early temporal patterning and defects in early temporal patterning transmit to late temporal defects including the time at which CB NBs stop divisions. Notch curtails expression of at least three early temporal factors (Cas, Svp, and Imp), suggesting that Notch may function broadly to close the early temporal window. It is known that the early to late temporal transition is dependent on both early intrinsic temporal factors, Cas and Svp, and on extrinsic steroid hormone signaling (ecdysone) (***Ren et al., 2017***; ***Syed et al., 2017***). Cas overexpression is sufficient to prolong Svp expression, but Cas is not required for Svp expression, and Svp primes CB NBs to respond to ecdysone (***Ren et al., 2017***). Whether Notch pathway activity curtails both Cas and Svp or just Cas remains an open question, however it has been reported that both Cas and Svp are associated with at least one enhancer that is responsive to Notch activity (***Zacharioudaki et al., 2016***). Also, it remains unknown whether Notch directly inhibits Imp or whether Notch indirectly inhibits Imp through Syp expression in response to ecdysone or a yet unidentified factor. Notch was recently shown to regulate timing of Sloppy-paired expression in the optic lobe (***Ray and Li, 2022***).

While some CB NBs maintained early Imp expression, others co-expressed both Imp and Syp, or expressed Syp alone. This suggests that Notch function is lineage-dependent and/or suggests that more than one pathway regulates lineage progression. While Cas is likely expressed in all CB NBs, Svp appears to be more restricted. Whether Notch inhibits early temporal progression only in Svp expressing CB NBs is not yet known. Somewhat unexpectedly, we also found a significant percentage of ectopically persisting CB NBs expressing the late temporal factor Syp. Syp promotes accumulation of nuclear Pros in CB NBs during pupal stages to induce terminal differentiation (***Maurange et al., 2008***; ***Yang et al., 2017***). This suggests that either a Pros-independent mechanism exists to eliminate CB NBs and/or that Syp regulates expression of additional unknown temporal factors required for Pros nuclear accumulation. As we report here, Syp and E93 inhibit expression and localization of Delta and decreased Delta leads to ectopic persistence of CB NBs. This is consistent with the notion that once CB NBs transition from early to late temporal factor expression in response to ecdysone, late temporal factors (Syp/E93) inhibit Delta. Whether this changes Notch activity and/or transcription of

Notch target genes is not yet known. Nevertheless, it will be important to identify the Notch transcriptional target genes that regulate lineage progression. One good place to start will be to follow up on the eight transcription factors we identified from data mining.

During early larval stages, CB NBs reactivate from quiescence and produce GMCs that express Delta (*Zacharioudaki et al., 2012*; *Sood et al., 2022*). This leads to the transactivation of Notch in CB NBs (*Sood et al., 2022*). The early temporal factor Imp positively regulates Delta in CB NBs and their GMC progeny. In developing egg chambers, Imp positively regulates Notch pathway activity by controlling Kuz localization (*Fic et al., 2019*). Whether Imp in CB NBs also regulates Kuz localization remains an open question. Delta is also expressed in cortex glia and regulates CB NB Notch activity. Whether Delta expression in cortex glia changes over time as is the case for CB NBs remains an open question. Hedgehog signaling in CB NBs promotes lineage progression downstream of Cas and Hedgehog ligands are produced in cortex glia and GMCs (*Chai et al., 2013*). Thus, CB NBs integrate cues from their GMC progeny and neighboring cortex glial cells to control temporal progression and lineage termination.

Although the exact course of temporal progression is yet to be defined in the mammalian nervous system, mammalian NSCs temporally express several factors including microRNAs, mRNA-binding proteins, and transcription factors, allowing them to produce deep layer neurons (early-born neural progeny), superficial layer neurons (late-born neural progeny), and glial cells sequentially throughout development (*Okano and Temple, 2009*; *MuhChyi et al., 2013*; *Oberst et al., 2019*; *Telley et al., 2019*). COUP-TFI and COUP-TFII, orthologs of *Drosophila* Svp, function as late temporal factors allowing NSCs to switch from producing early-born neural fates to late-born neural fates (*Naka et al., 2008*). This is similar to the function of Svp in the *Drosophila* brain where Svp mutants failed to switch from early Chinmo positive daughters to late Broad-complex positive daughters (*Maurange et al., 2008*). Similarly, mammalian NSCs temporally express RNA-binding protein Imp-1, ortholog of *Drosophila* Imp, and in Imp-1 deficient animals, NSCs are lost prematurely similar to premature loss of CB NBs seen in *Drosophila* (*Nishino et al., 2013*; *Yang et al., 2017*; *Pahl et al., 2019*). As in *Drosophila* NBs, temporal progression of mammalian NSCs is not completely dependent on cell-intrinsic cues but also requires cell-extrinsic cues like feedback signals from the lineage and environmental cues (*Okamoto et al., 2016*; *Oberst et al., 2019*; *Zhang et al., 2020*), however, very little is known about the signaling pathways regulating the transitions from early to late temporal fates. Even though Notch activity is required for the temporal switch from neurogenesis to gliogenesis in mammalian NSCs, it remains unclear whether Notch function extends to regulation of temporal progression required for the switch from early-born to late-born neuron subtypes (*Morrison et al., 2000*; *Grandbarbe et al., 2003*; *Ohtsuka and Kageyama, 2019*).

# Materials and methods

Key resources table

| Reagent type (species) or resource | Designation | Source or reference | Identifiers | Additional information |
|---|---|---|---|---|
| Antibody | Anti-Dpn (rat monoclonal) | Abcam | ab195173 | IF (1:1000) |
| Antibody | Anti-GFP (chicken polyclonal) | Abcam | ab13970 | IF (1:500) |
| Antibody | Anti-dsRed (rabbit polyclonal) | Clontech | 632496 | IF (1:1000) |
| Antibody | Anti-PH3 (rabbit polyclonal) | Millipore | 06-570 | IF (1:1000) |
| Antibody | Anti-Repo (mouse monoclonal) | Developmental Studies Hybridoma Bank | 8D12 | IF (1:5) |
| Antibody | Anti-Prospero (mouse monoclonal) | Developmental Studies Hybridoma Bank | MR1A | IF (1:1000) |
| Antibody | Anti-Svp (mouse monoclonal) | Developmental Studies Hybridoma Bank | 5B11 | IF (1:10) |
| Antibody | Anti-Dlg (mouse monoclonal) | Developmental Studies Hybridoma Bank | 4F3 | IF (1:40) |
| Antibody | Anti-Scribble (rabbit polyclonal) | Gift from Chris Q Doe | | IF (1:500) |

*Continued on next page*

*Continued*

| Reagent type (species) or resource | Designation | Source or reference | Identifiers | Additional information |
|---|---|---|---|---|
| Antibody | Anti-Cas (rabbit polyclonal) | Gift from Chris Q Doe | | IF (1:500) |
| Antibody | Anti-Syp (rabbit polyclonal) | Gift from Chris Q Doe | | IF (1:250) |
| Antibody | Anti-E93 (guinea pig polyclonal) | Gift from Chris Q Doe | | IF (1:250) |
| Antibody | Anti-Imp (rabbit polyclonal) | Gift from Paul MacDonald | | IF (1:250) |
| Antibody | Anti-Imp (rat polyclonal) | Gift frrom Claude Desplan | | IF (1:250) |
| Antibody | Anti-Dpn (guinea pig polyclonal) | Gift from Claude Desplan | | IF (1:1000) |
| Antibody | Alexa 488 (goat anti-chicken polyclonal) | Thermo Fisher Scientific | A32931 | IF (1:300) |
| Antibody | Alexa 555 (goat anti-rat polyclonal) | Thermo Fisher Scientific | A48263 | IF (1:300) |
| Antibody | Alexa 647 (goat anti-rat polyclonal) | Thermo Fisher Scientific | A48265 | IF (1:300) |
| Antibody | Alexa 405 (goat anti-rabbit polyclonal) | Thermo Fisher Scientific | A48254 | IF (1:300) |
| Antibody | Alexa 555 (goat anti-rabbit polyclonal) | Thermo Fisher Scientific | A21428 | IF (1:300) |
| Antibody | Alexa 633 (goat anti-rabbit polyclonal) | Thermo Fisher Scientific | A21071 | IF (1:300) |
| Antibody | Alexa 405 (goat anti-mouse polyclonal) | Thermo Fisher Scientific | A48255 | IF (1:300) |
| Antibody | Alexa 488 (goat anti-mouse polyclonal) | Thermo Fisher Scientific | A11001 | IF (1:300) |
| Antibody | Alexa 555 (goat anti-mouse polyclonal) | Thermo Fisher Scientific | A32727 | IF (1:300) |
| Antibody | Alexa 488 (goat anti-guinea pig polyclonal) | Thermo Fisher Scientific | A11073 | IF (1:300) |
| Antibody | Alexa 555 (goat anti-guinea pig polyclonal) | Thermo Fisher Scientific | A21435 | IF (1:300) |
| Chemical compound, drug | SlowFade Diamond antifade reagent | Invitrogen | Catalog # S36963 | |
| Chemical compound, drug | SlowFade Gold antifade reagent | Invitrogen | Catalog # S36937 | |
| Chemical compound, drug | Normal Goat Serum | Thermo Fisher Scientific | Catalog # 31873 | |
| Chemical compound, drug | Paraformaldehyde 16% solution EM grade | Electron Microscopy Sciences | Catalog # 15710 | |
| Chemical compound, drug | Schneider's *Drosophila* media | Gibco | Catalog # 21720-024 | |
| Chemical compound, drug | Triton X-100 | Sigma | Catalog # T9284 | |
| Software, algorithm | ImageJ/Fiji | Fiji | | http://fiji.sc/ |
| Software, algorithm | LAS AF | Leica Microsystems | | https://www.leica-microsystems.com/products/microscope-software/details/product/leica-las-x-ls/ |
| Software, algorithm | Prism 9 | GraphPad | | https://www.graphpad.com/scientific-software/prism/ |
| Software, algorithm | Photoshop 2022 | Adobe | | https://www.adobe.com/products/photoshop.html |

*Continued on next page*

*Continued*

| Reagent type (species) or resource | Designation | Source or reference | Identifiers | Additional information |
|---|---|---|---|---|
| Software, algorithm | Illustrator 2022 | Adobe | | https://www.adobe.com/products/illustrator.html |
| Software, algorithm | R-studio | R-studio | | https://www.rstudio.com/ |
| Genetic reagent (*D. melanogaster*) | Oregon R | Bloomington Drosophila Stock Center | 5 | |
| Genetic reagent (*D. melanogaster*) | wor-Gal4 | *Albertson and Doe, 2003* | | |
| Genetic reagent (*D. melanogaster*) | tubulin-Gal80(ts) | Bloomington Drosophila Stock Center | 7108 | |
| Genetic reagent (*D. melanogaster*) | repo-Gal4 | Bloomington Drosophila Stock Center | 7415 | |
| Genetic reagent (*D. melanogaster*) | NP0577-Gal4 | Kyoto Stock Center | 112228 | |
| Genetic reagent (*D. melanogaster*) | repo-Gal80 | *Awasaki et al., 2008* | | |
| Genetic reagent (*D. melanogaster*) | UAS-Notch RNAi (HMS00001) | Bloomington Drosophila Stock Center | 33611 | |
| Genetic reagent (*D. melanogaster*) | UAS-Kuzbanian RNAi (HMS05424) | Bloomington Drosophila Stock Center | 66958 | |
| Genetic reagent (*D. melanogaster*) | UAS-Su(H)RNAi (HMS05748) | Bloomington Drosophila Stock Center | 67928 | |
| Genetic reagent (*D. melanogaster*) | UAS-Delta RNAi (HMS01309) | Bloomington Drosophila Stock Center | 34322 | |
| Genetic reagent (*D. melanogaster*) | UAS-Serrate RNAi (HMS01179) | Bloomington Drosophila Stock Center | 34700 | |
| Genetic reagent (*D. melanogaster*) | UAS-dp110 | Bloomington Drosophila Stock Center | 25914 | |
| Genetic reagent (*D. melanogaster*) | Delta-GFP | Bloomington Drosophila Stock Center | 59819 | |
| Genetic reagent (*D. melanogaster*) | Serrate-GFP | Bloomington Drosophila Stock Center | 59824 | |
| Genetic reagent (*D. melanogaster*) | pcna-GFP | *Thacker et al., 2003* | | |
| Genetic reagent (*D. melanogaster*) | E(spl)mg-GFP | *Almeida and Bray, 2005* | | |
| Genetic reagent (*D. melanogaster*) | Imp-GFP | Bloomington Drosophila Stock Center | 60237 | |
| Genetic reagent (*D. melanogaster*) | UAS-Imp RNAi (HMS01168) | Bloomington Drosophila Stock Center | 34977 | |
| Genetic reagent (*D. melanogaster*) | UAS-Syp-RB-HA | Gift from Tzumin Lee | | |
| Genetic reagent (*D. melanogaster*) | UAS-Eip93F WT | Zurich FlyORF | F000587 | |
| Genetic reagent (*D. melanogaster*) | hsFlp (on X) | Gift from Iswar Hariharan | | |
| Genetic reagent (*D. melanogaster*) | Act5c-FRT-CD2-FRT-Gal4, UAS-RFP | Bloomington Drosophila Stock Center | 30558 | |

*Continued on next page*

*Continued*

| Reagent type (species) or resource | Designation | Source or reference | Identifiers | Additional information |
|---|---|---|---|---|
| Genetic reagent (*D. melanogaster*) | Act5c-FRT-CD2-FRT-Gal4, UAS-GFP | Gift from Iswar Hariharan | | |
| Genetic reagent (*D. melanogaster*) | UAS-mCD8-mRFP | Bloomington Drosophila Stock Center | 27399 | |
| Genetic reagent (*D. melanogaster*) | UAS-mCD8-mGFP | Bloomington Drosophila Stock Center | 5137 | |
| Genetic reagent (*D. melanogaster*) | Notch55e11 FRT19A | Bloomington Drosophila Stock Center | 28813 | |
| Genetic reagent (*D. melanogaster*) | hsflp, tubgal80, FRT19A; tubGal4, UASmCD8GFP | Gift from Ben Ohlstein | | |

## Fly stocks

Fly stocks used in this study and their source are listed in the Key resources table.

## Animal husbandry

All animals were raised in uncrowded conditions at 25°C with the exception of animals with tub-GAL80(ts). For experiments using *tub-Gal80^{ts}* (temperature-sensitive), animals were kept in uncrowded conditions at 29°C and dissected at developmental timings to be equivalent to development at 25°C unless otherwise stated. Animals were staged from hatching for larval dissections and from white prepupae for pupal dissections.

## Induction of clones

For induction of Flp-FRT and MARCM clones, animals were heat shocked at 37°C between 30 and 60 min at L1 and dissected at the stated developmental timings.

## Temperature shift experimental paradigm

For early knockdown experiments, animals were raised at 29°C until 72 hr ALH (equivalent to 25°C) and then moved to 18°C to develop until 48 hr APF (equivalent to 25°C). For late knockdown experiments, animals were raised at 18°C until 72 hr ALH (equivalent to 25°C) and then moved to 29°C to develop until 48 hr APF (equivalent to 25°C).

## Immunofluorescence and confocal imaging

Larval, pupal, and adult brains were dissected as previously described (*Pahl et al., 2019*). In brief, dissected tissues were fixed in 4% EM grade formaldehyde in PEM buffer for 20 min (larvae) or 30 min (pupae and adults) and rinsed in 1× PBS with 0.1% Triton X-100 (PBT). Tissues were blocked overnight at 4°C in 10% normal goat serum in PBT followed by antibody staining. Primary antibodies used are listed in the Key resources table. To detect primary antibodies, Alexa Fluor-conjugated secondary antibodies (Thermo Fisher) listed in the Key resources table were used. Images encompassing the entire brain hemispheres were acquired using a Leica SP8 laser scanning confocal microscope equipped with a 63×/1.4 NA and 40×/1.3 NA oil immersion objectives and analyzed using Fiji software. All images were processed using Fiji and Adobe Photoshop software and figures assembled using Adobe Illustrator software. NBs were identified based on Dpn expression and superficial location. The Fiji 'cell counter' plugin was used to count and track number of Dpn positive NBs. NB size was calculated by averaging the lengths of two perpendicular lines through the center of the NB in Fiji. Quantification of fluorescence was performed in Fiji. Nuclear E93 levels were quantified as follows: CB NB nuclei labeled with Dpn were manually traced and the average E93 fluorescence intensity measured in the nucleus. Normalized E93 nuclear fluorescence intensity was determined as a ratio of nuclear E93 fluorescence intensity in a clone NB to nuclear E93 fluorescence intensity in a control NB in the same z-plane. All data is represented as mean ± standard error of the mean and statistical significance was determined using unpaired two-tailed Student's t-tests or ANOVAs in Prism 9.

## RNA-sequence data analysis

Differential gene expression (DGE) analysis was performed using the publicly available dataset published in *Liu et al., 2015* (GSE71103). We aligned FASTQ data to a reference Dm genome using STAR (version 2.7.2b) (*Dobin et al., 2013*). Read count tables were generated using the htseq-count package from bioconda, and DGE was performed using DESeq2 (*Love et al., 2014*). We made contrasts between 24 hr ALH and 84 hr ALH antennal lobe NBs in order to identify DEGs of interest (adj. p-value <0.1 and log fold change >1). The list of 1861 DEGs was compared to the list of downstream genetic interactions of Notch pathway from FlyBase (https://flybase.org/reports/FBgn0004647) to obtain the list of 59 DEGs of interest. Finally, VST (variance-stabilizing transformation) normalized counts of the 59 genes were used to generate the heatmaps using heatmap.2 from the ggplot2 package (https://ggplot2.tidyverse.org). To obtain common target genes between Notch and Imp, we compared our list of 59 genes to the top 1000 Imp target genes obtained via RIP-seq by *Samuels et al., 2020*. We performed similar comparison on the 274 Syp target genes obtained via RIP-seq by *McDermott et al., 2014*, and our list of 59 genes to obtain common Notch and Syp target genes.

## Acknowledgements

We thank Claude Desplan, Chris Doe, Cheng-Yu Lee, Tzumin Lee, Ben Ohlstein, Bloomington Drosophila Stock Center, Harvard TRiP, and the Developmental Studies Hybridoma Bank for providing flies and antibody reagents. We thank all Siegrist lab members for providing helpful comments and suggestions on the manuscript and research project.

## Additional information

### Funding

| Funder | Grant reference number | Author |
| --- | --- | --- |
| National Institute of Health | R35GM141886 | Sarah E Siegrist |

The funders had no role in study design, data collection and interpretation, or the decision to submit the work for publication.

### Author contributions

Chhavi Sood, Conceptualization, Data curation, Formal analysis, Validation, Investigation, Visualization, Methodology, Writing - original draft, Writing – review and editing; Md Ausrafuggaman Nahid, Investigation, Methodology; Kendall R Branham, Formal analysis, Investigation; Matt Pahl, Susan E Doyle, Investigation; Sarah E Siegrist, Conceptualization, Resources, Supervision, Funding acquisition, Project administration, Writing – review and editing

### Author ORCIDs

Sarah E Siegrist ⬤ https://orcid.org/0000-0003-0685-5387

Reviewer #1 (Public Review): https://doi.org/10.7554/eLife.88565.3.sa1
Reviewer #2 (Public Review): https://doi.org/10.7554/eLife.88565.3.sa2
Reviewer #3 (Public Review): https://doi.org/10.7554/eLife.88565.3.sa3
Author Response https://doi.org/10.7554/eLife.88565.3.sa4

## Additional files

### Supplementary files

• Supplementary file 1. Panel genotypes. Table of genotypes listed by panel of each figure and figure supplement.

• MDAR checklist

## Data availability

All data generated or analysed during this study are included in the manuscript and supporting files.

The following previously published dataset was used:

| Author(s) | Year | Dataset title | Dataset URL | Database and Identifier |
|---|---|---|---|---|
| Sugino K, Lee T, Liu Z, Yang C | 2015 | Opposite Imp/Syp temporal gradients govern birth time-dependent neuronal fates | https://www.ncbi.nlm.nih.gov/geo/query/acc.cgi?acc=GSE71103 | NCBI Gene Expression Omnibus, GSE71103 |

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
