## [Editor Report · eLife assessment]

This **useful** study reports on how Notch activity regulates the termination of neurogenesis in central brain during larval-pupal stages in *Drosophila*. The evidence supporting the claims is **solid**. The work will be of interest to developmental neurobiologists.

---

## [Referee Report · Reviewer #1 (Public Review)]

In this manuscript, the authors are building on their previous work showing Delta-Notch regulates the entrance and exit from embryo-larval quiescence of neural stem cells of the central brain (called CB neuroblasts (NB) (PMID: 35112131)). Here they show that continuous depletion of Notch in NBs from early embryogenesis leads to cycling NBs in the adult. This - cycling NBs in the adult - is not seen in controls. The assumption here is that these Notch-RNAi NBs in adults are those that did not undergo terminal differentiation in pupal development. The authors show that Notch is activated by its ligand Delta which is expressed on the GMC daughter cell and on cortex glia. They determine that the temporal requirement for Notch activity is 0-72 hours after larval hatching (ALH) (i.e., 1st instar through mid-3rd instar at 25C). In NBs/GMCs depleted for Notch, early temporal markers were still expressed at time points when they should be off and late markers were delayed in expression. These effects were observed in ~20-40% of NBs (Figures 5 and 6). Through mining existing data sets, they found that the early temporal factor Imp - an RNA binding protein - can bind Delta mRNA. They show that Delta transcripts decrease over time, leading to the hypothesis that Delta mRNA is repressed by the late temporal factors. Over-expressing late factors Syp or E93 earlier in development leads to downregulation of a Delta::GFP protein trap. These results lead to a model in which Notch regulates expression of early temporal factors and early temporal factors regulate Notch activity through translation of Delta mRNA.

There are several strengths of this study and no major weaknesses. The authors report rigorous measurements and statistical analyses throughout the study. Their conclusions are appropriate for the results. Data mining revealed an important mechanism - that Imp binds Delta mRNA - supporting the model that that early temporal factors promote Delta expression, which in turn promotes Notch signaling.

An appraisal: The authors use temperature shifts with Gal80TS to show that Notch is required between 0-72 hours ALH. They show with the use of known markers of the temporal factors and Delta protein trap, that Imp promotes Delta protein expression and the later temporal factors reduce Delta, although the molecular mechanisms are not clearly delineated. Overall, these data support their model that the reduction of Delta expression during larval development leads to a loss of Notch activity.

As noted in the Discussion, this study raises many questions about what Notch does in larval CB NBs. For example, does it inhibit Castor or Imp? Is Notch required in certain neural lineages and not others. These studies will be of interest in the community of developmental neurobiologists.

---

## [Referee Report · Reviewer #2 (Public Review)]

As I indicated in the initial review, the experiments are well conceived and executed, and the data are clear. I also agree with the authors that this work represents a key first step toward understanding how Notch signaling contributes to temporal control of fly neuroblasts. It is my opinion that the authors fall short of demonstrating how Notch signaling and temporal identity genes at the chromatin levels. I find this disappointing given the availability of various tools for looking at dynamic regulation of gene activity at high resolution. Given these weaknesses, my opinion is that the study is descriptive and lacks mechanistic explanation.

---

## [Referee Report · Reviewer #3 (Public Review)]

In this study, the authors investigate the effects of Notch pathway inactivation on the termination of *Drosophila* neuroblasts at the end of development. They find that termination is delayed, while temporal patterning progression is slowed down. Forcing temporal patterning progression in a Notch pathway mutant restores correct timing of neuroblast elimination. Finally they show that Imp, an early temporal patterning factor promotes Delta expression in neuroblast lineages. This indicates that feedback loops between temporal patterning and lineage-intrinsic Notch activity fine tunes timing of early to late temporal transitions and is important to schedule NB termination at the end of development.

The study adds another layer of regulation that finetunes temporal progression in *Drosophila* neural stem cells. This mechanism appears to be mainly lineage intrinsic - Delta being expressed from NBs and their progeny, but also partly niche-mediated - Delta being also expressed in glia but with a minor influence. Together with a recent study (PMID: 36040415), this work suggests that Notch signaling is a key player in promoting temporal progression in various temporal patterning system. As such it is of broad interest for the neuro-developmental community.

Strengths

The data are based on genetic experiments which are clearly described and mostly convincing. The study is interesting, adding another layer of regulation that finetunes temporal progression in *Drosophila* neural stem cells. This mechanism appears to be mainly lineage intrinsic - Delta being expressed from NBs and their progeny, but also partly niche-mediated - Delta being also expressed in glia but with a minor influence. A similar mechanism has been recently described, although in a different temporal patterning system (medulla neuroblasts of the optic lobe - PMID: 36040415). It is overall of broad interest for the neuro-developmental community.

Weaknesses

The mechanisms by which Notch signaling regulates temporal patterning progression are not investigated in details. For example, it is not clear whether Notch signaling directly regulates temporal patterning genes, or whether the phenotypes observed are indirect (for example through the regulation of the cell-cycle speed). The authors could have investigated whether temporal patterning genes are directly regulated by the Notch pathway via ChIP-seq of Su(H) or the identification of potential binding sites for Su(H) in enhancers. A similar approach has been recently undertaken by the lab of Dr Xin Li, to show that Notch signaling regulates sequential expression of temporal patterning factors in optic lobes neuroblasts (PMID: 36040415), which exhibit a different temporal patterning system than central brain neuroblasts in the present study. As such, the mechanistic insights of the study are limited.

---

## [Author Response]

The following is the authors’ response to the original reviews.

We very much appreciate the constructive comments provided by the reviewers. We have incorporated many of their suggestions and believe the manuscript is much improved.

In brief, we updated the text as suggested and have included three additional panels in supplementary fig. S2E-G. This additional data provides further support that the ectopically persisting neuroblasts are actively dividing and that cell cycle defects alone do not account for temporal patterning phenotypes.

**Reviewer #1 (Public Review):**
In this manuscript, the authors are building on their previous work showing Delta-Notch regulates the entrance and exit from embryo-larval quiescence of neural stem cells of the central brain (called CB neuroblasts (NB) (PMID: 35112131)). Here they show that continuous depletion of Notch in NBs from early embryogenesis leads to cycling NBs in the adult. This - cycling NBs in the adult - is not seen in controls. The assumption here is that these Notch-RNAi NBs in adults are those that did not undergo terminal differentiation in pupal development. The authors show that Notch is activated by its ligand Delta which is expressed on the GMC daughter cell and on cortex glia. They determine that the temporal requirement for Notch activity is 0-72 hours after larval hatching (ALH) (i.e., 1st instar through mid-3rd instar at 25C). In NBs/GMCs depleted for Notch, early temporal markers were still expressed at time points when they should be off and late markers were delayed in expression. These effects were observed in ~20-40% of NBs (Figures 5 and 6). Through mining existing data sets, they found that the early temporal factor Imp - an RNA binding protein - can bind Delta mRNA. They state that Delta transcripts decrease over time (without any reference to a Figure or to published work), leading to the hypothesis that Delta mRNA is repressed by the late temporal factors. Over-expressing late factors Syp or E93 earlier in development leads to downregulation of a Delta::GFP protein trap. These results lead to a model in which Notch regulates expression of early temporal factors and early temporal factors regulate Notch activity through translation of Delta mRNA.There are several strengths of this study. The authors report rigorous measurements and statistical analyses throughout the study. Their conclusions are appropriate for the results. Data mining revealed an important mechanism - that Imp binds Delta mRNA - supporting the model that early temporal factors promote Delta expression, which in turn promotes Notch signaling.There are also several weaknesses:1. The activation of Notch in NBs by Delta in GMCs was already shown by this group in their Dev 2022 paper, reducing some of the impact of this study.

In our previous work, we reported that Delta-expressing GMCs transactivate Notch in neuroblasts during the embryonic to larval transition. In the current manuscript, we show that Delta is expressed in GMCs and cortex glia and both sources transactivate Notch in neuroblasts during later developmental stages. This is in agreement with work published by others and while not novel per se, is a necessary first step for understanding which neighboring cell types control Notch pathway activity. During the embryonic to larval transition, glia do not contribute likely because they have not yet grown to ensheath CB NBs and their recently born progeny.

1. The authors do not explain their current results in context of their prior paper (2022 Dev) until the Discussion, but this would be useful to read in the Introduction. Similarly, it would be good to mention that in the 2022 paper, they find a significant number of wor>Notch RNAi NBs at 2 AHL that are cycling. Are the adult Notch RNAi in this study descended from those NBs at 2 hours ALH in the 2022 study? In other words, how does the early requirement for Notch between 0-72 hours ALH reported in the current study relate to the Notch-depleted NBs identified in the 2022 paper?

We have now included the following text in the intro: “We recently reported that Notch signaling regulates CB NB quiescence during the embryonic to larval transition (Sood et al., 2022). When Notch is knocked down, some CB NBs continue dividing during this transition. We also reported that Notch activity becomes attenuated in quiescent CB NBs because CB NBs are no longer dividing and producing Delta-expressing GMC daughters for Notch pathway transactivation. Moreover, low Notch is necessary for CB NBs to reactivate from quiescence in response to dietary nutrients (Sood et al., 2022).

Here we report that Notch signaling also regulates neurogenesis termination during pupal stages. When Notch is knocked down, CB NBs maintain early temporal factor expression longer resulting in a delay of late temporal factor expression with prolonged neurogenesis into late pupal stages and early adulthood. This defect in temporal patterning (switching from early to late) occurs after CB NB exit from quiescence suggesting that Notch is required at multiple times throughout development in controlling CB NB proliferation decisions.”

We do not know whether the neuroblasts that fail to enter quiescence are the same that fail to terminate divisions during pupal stages, however there are many more that fail to terminate divisions during pupal stages.

1. Most of the experiments rely upon continuous depletion of Notch from embryonic stage 8 until adulthood using the wor-GAL4 driver. There is no lineage tracing of this driver and there is no citation about the published expression pattern of this driver. The inclusion of these details is important for a broad audience journal.

The reference for the driver is included in supplementary data, under the heading “Experimental model:*Drosophila melanogaster*”. This GAL4 driver is widely used and one of the most accepted in the field.

1. Most of the experiments utilize a single RNAi transgene for Notch, Delta, Imp, Syp, E93. There are no experiments demonstrating the efficacy of the RNAi lines and no references to prior use and/or efficacy of these lines.

All RNAi lines used in these studies have been published previously, by our group as well as others and sources for the lines are listed in supplementary data, under the heading “Experimental model:*Drosophila melanogaster*”. Efficiency of these lines have been verified using antibody labeling (data not shown) and by assaying activity of Notch activity reporters (shown in Fig. 2).

An appraisal: The authors use temperature shifts with Gal80TS to show that Notch is required between 0-72 hours ALH. They show with the use of known markers of the temporal factors and Delta protein trap, that Imp promotes Delta protein expression and the later temporal factors reduce Delta, although the molecular mechanisms are not clearly delineated. Overall, these data support their model that the reduction of Delta expression during larval development leads to a loss of Notch activity.As noted in the Discussion, this study raises many questions about what Notch does in larval CB NBs. For example, does it inhibit Castor or Imp? Is Notch required in certain neural lineages and not others. These studies will be of interest in the community of developmental neurobiologists.
**Reviewer #2 (Public Review):**
Embryonic stem cells extensively proliferate to generate the necessary number of cells that are required for organogenesis, and their proliferation must be timely terminated to allow for proper patterning. Thus, timely termination of stem cell proliferation is critical for proper development. Numerous studies have suggested that cell-extrinsic changes in the surrounding niche environment drive the termination of stem cell proliferation. By contrast, cell-intrinsic mechanisms that terminate stem cell proliferation remain poorly understood. Fruit fly larval brain neuroblasts provide an excellent model for mechanistic investigation of intrinsic control of stem cell proliferation due to the wealth of information on molecular marks, gene functions and lineage hierarchy. Sood et al. conducted a genetic screen to identify genes that are required for the termination of neuroblast proliferation in metamorphosis and found that Notch and its ligand Delta contribute to their exit from cell cycle. They showed that knocking down Notch or delta function in larval neuroblasts allows them to persist into adulthood and remain proliferative when no neuroblasts can be detected in wild-type adult brains. By carrying out a well-designed temperature-shift experiment, the authors showed that Notch is required early during larval development to promote timely exit from cell cycle in metamorphosis. The authors went on to show that attenuating Notch signaling prolongs the expression of temporal identity genes castor and seven-up perturbing the switch from Imp to Syp/E93. Finally, they showed that knocking down Imp function or overexpressing E93 can restore the elimination of neuroblasts in Notch/delta mutant brains.Overall, the experiments are well conceived and executed, and the data are clear. However, the data reported in this study represent incremental progress in improving our mechanistic understanding of the termination of neuroblast proliferation.

We respectfully disagree with this statement. Because Notch signaling is implicated in neurogenesis termination and Notch activity is regulated by GMCs and glia, it strongly suggests that NB proliferation and timing cues are controlled in a non-autonomous manner through direct interactions with NBs and their neighbors. This is in contrast to temporal patterning during embryogenesis which is largely believed to be controlled NB-autonomously. In addition, to our knowledge, no one has yet reported that CB NBs fail to terminate cell divisions on time when Notch activity is reduced during normal development. In fact, reported NB phenotypes associated with Notch loss of function have been surprisingly subtle until now.

Some of the data seem to represent more careful analyses of previously published observations described in the Zacharioudaki et al., Development 2016 paper while others seem to contradict to the results in this study.

The Zacharioudaki et al., Development 2016 paper is terrific. One key difference between our work and theirs, is that we look at Notch pathway knockdown and loss of function phenotypes, whereas in the Zacharioudaki 2016 paper, the authors report phenotypes associated with Notch constitutive activation. It has been known for some time that constitutively active Notch leads to tumorigenic phenotypes particularly in type II lineages. Zacharioudaki and colleagues further determined that some of the classically known temporal transcription factors were ectopically expressed in these stem cell tumors.Here we show that under normal developmental conditions, Notch pathway activity controls CB NB temporal patterning.

Gaultier et al., Sci. Adv. 2022 suggested that Grainyhead is required for the termination of neuroblast proliferation in a neuroblast tumor model, and grainyhead is a direct target of Notch signaling. Thus, Grainyhead should be a key downstream effector of Notch signaling in terminating castor and seven-up expression. Identical to Notch signaling, Grainyhead is also expressed through larval development. Grainyhead can function as a classical transcription factor as well as a pioneer factor raising the possibility that temporal regulation of neurogenic enhancer accessibility might be at play in allowing Notch signaling in early larval development to set up termination of castor and seven-up expression in metamorphosis. Diving deeper into how dynamic changes in chromatin in neurogenic enhancers affect the termination of neuroblast proliferation will significantly improve our understanding of termination of stem cell proliferation in diverse developing tissue.
**Reviewer #3 (Public Review):**
In this study, the authors investigate the effects of Notch pathway inactivation on the termination of *Drosophila* neuroblasts at the end of development. They find that termination is delayed, while temporal patterning progression is slowed down. Forcing temporal patterning progression in a Notch pathway mutant restores the correct timing of neuroblast elimination. Finally, they show that Imp, an early temporal patterning factor promotes Delta expression in neuroblast lineages. This indicates that feedback loops between temporal patterning and lineage-intrinsic Notch activity fine tunes timing of early to late temporal transitions and is important to schedule NB termination at the end of development.The study adds another layer of regulation that finetunes temporal progression in *Drosophila* neural stem cells. This mechanism appears to be mainly lineage intrinsic - Delta being expressed from NBs and their progeny, but also partly niche-mediated - Delta being also expressed in glia but with a minor influence. Together with a recent study (PMID: 36040415), this work suggests that Notch signaling is a key player in promoting temporal progression in various temporal patterning system. As such it is of broad interest for the neuro-developmental community.StrengthsThe data are based on genetic experiments which are clearly described and mostly convincing. The study is interesting, adding another layer of regulation that finetunes temporal progression in *Drosophila* neural stem cells. This mechanism appears to be mainly lineage intrinsic - Delta being expressed from NBs and their progeny, but also partly niche-mediated - Delta being also expressed in glia but with a minor influence. A similar mechanism has been recently described, although in a different temporal patterning system (medulla neuroblasts of the optic lobe - PMID: 36040415). It is overall of broad interest for the neuro-developmental community.WeaknessesThe mechanisms by which Notch signaling regulates temporal patterning progression are not investigated in details. For example, it is not clear whether Notch signaling directly regulates temporal patterning genes, or whether the phenotypes observed are indirect (for example through the regulation of the cell-cycle speed). The authors could have investigated whether temporal patterning genes are directly regulated by the Notch pathway via ChIP-seq of Su(H) or the identification of potential binding sites for Su(H) in enhancers.

This is already known for svp and cas and we have now included this information in the discussion.Thank you.

“Whether Notch pathway activity curtails both Cas and Svp or just Cas remains an open question, however it has been reported that both cas and svp are associated with at least one enhancer that is responsive to Notch activity (Zacharioudaki et al., 2016).”

A similar approach has been recently undertaken by the lab of Dr Xin Li, to show that Notch signaling regulates sequential expression of temporal patterning factors in optic lobes neuroblasts (PMID: 36040415), which exhibit a different temporal patterning system than central brain neuroblasts in the present study. As such, the mechanistic insights of the study are limited.
**Reviewer #1 (Recommendations For The Authors):**
1. There are missing controlsa) Fig. 1F and Fig. 6A - The authors should generate and show images of control clones (FRT19A) stained with the same markers as Notch clones.

Fig. 1F is at 48 hours APF. In control clones, there are no Dpn positive cells present, as stated in the text and therefore no confocal images are shown. Same for Fig. 6A, there are no Dpn positive cells in control clones in the brain at this time, therefore nothing to double label.

1. This result is incorrectly described in the Resultsa) P. 5 "Ectopically persisting N RNAi CB NBs expressed the NB transcription factor Deadpan (Dpn), the S-phase indicator pcnaGFP, and were small on average, similar in size to control CB NBs at earlier pupal stages (Fig. 1B,C,E)." The Notch RNAi NBs were larger (not smaller) than controls in Fig. 1E at 30, 48, 72 h APF and in adults.

Thank you for this comment. We have changed the language in the main text as follows:

“Ectopically persisting N RNAi CB NBs (CB NBs at 48 hours APF and beyond) expressed the NB transcription factor Deadpan (Dpn), the S-phase indicator pcnaGFP, and were small on average compared to control CB NBs during earlier developmental stages (L3 control, average diameter10-15μms) (Fig. 1B,C,E). However, at 30 hours APF when control CB NBs are still present, N RNAiCB NBs were larger on average (Fig. 1B,C,E).”

1. This sentence needs clarification/editinga) P. 4: " Independent of neurogenesis timing and the mechanism by which CB NB stop divisions, temporal patterning plays a key role". A key role in what?

Thank you again. We have changed the text to the following:

“Independent of neurogenesis timing and the mechanism by which CB NB stop divisions, temporal patterning plays a key role in controlling numbers and types of neurons made within each of the NB lineages (Maurange et al., 2008; Tsuji et al., 2008; Bahrampour et al., 2017; Yang et al., 2017; Pahl et al., 2019).”

1. Some sentences need references or data to support them.a) P. 9 Please provide a reference to support the statement that Delta is a known Notch target

We have included a reference.

b) P. 9 - please provide a reference or data to support the statement that Delta transcripts decrease over time in larval CB NBs.

This result is shown in Fig. 7B.

1. Fig. 7A - it is difficult to appreciate the purple highlighting.

We have changed the colors as suggested.

**Reviewer #2 (Recommendations For The Authors):**
1. In Fig. 4C, why does late knockdown of delta lead to ectopic persistence of NBs but late knockdown of Notch has no effect?

This could be due to many things including differences in efficiency of UAS-RNAi lines. The point is that Delta/Notch is required early, but not late. Although some DeltaRNAi CB NBs are still present, the number compared to 48 hours APF is greatly reduced.

1. It is surprising that Delta expression in NBs/GMCs appears to play a more important role in activating Notch signaling in neuroblasts than Delta expression in cortex glia. Please explain how Delta can cell autonomously activate Notch signaling.

We are not proposing that Delta activates Notch cell autonomously, but are proposing that Delta inGMCs transactivates Notch in NBs. After NBs divide Delta is partitioned to GMCs. Quiescent NBs have low to no Notch pathway activity, likely because they are not producing Delta expressing GMC daughters (Sood, 2022).

Please also reconcile the difference in gene expression induced by delta[RNAi] in this study and the delta-mutant allele used in the Zacharioudaki et al study.

We are unsure what the reviewer is asking here and therefore can not reconcile any differences in gene expression between the dlRNAi line and the mutant allele. What gene expression needs to be reconciled? Zacharioudaki is listed as first author on four manuscripts. Which paper is being referred to?

1. In Fig. 2J-L, why does knocking down delta in glia lead to loss of Scrib expression in neuroblasts and their surrounding progeny?

We are not sure if it does or not. We only use Scrib as a membrane marker to identify and locate cells and neuropil regions of interest.

1. The phrase "Notch is active early" is misleading when multiple labs have shown that Notch signaling is active in neuroblasts throughout larval development.

Good point! We have rewritten the statement: “Somewhat paradoxically, we find that early Notch activity is required to terminate CB NB divisions late.”

1. Neuroblasts that persist into adulthood are "smaller and Dpn-positive/PCNA-GFP-positive". Are they really neuroblasts? Can the authors verify the identity of these "persistent neuroblasts" with other molecular markers as well as functional assessment by inducing lineage clones?

We have no doubt that these cells are NBs. Because we examine brains over time, these cells can be tracked using the markers, Scrib, Dpn, and pcna. These cells also undergo asymmetric cell division (Refer to Fig. S2F) and express other markers characteristic of CB NBs (mir and insc-not shown). We have made clones and see the same phenotype (ectopic persistence) in both MARCM clones and in “flip-out” clones.

**Reviewer #3 (Recommendations For The Authors):**
I have a few issues that need to be addressed to reinforce some of the conclusions:1. It is unclear whether NBs that persist in late pupal or adult stages have just failed to differentiate or whether they continue to divide, leading to supernumerary progeny (as shown for NBs that are stalled in temporal patterning like in svp mutant NBs (Maurange et al. 2008)). EdU or PH3 staining could be done in adults to clarify this point.

In this manuscript, we make use of pcna:GFP, a reporter for E2F activity as an indicator of cell proliferation. We certainly observe Dpn positive cells that only weakly express the reporter, suggesting that these cells are not actively dividing or dividing at a reduced rate. However, by far most of the ectopically persisting CB NBs strongly express the reporter and generate pcnaGFP expressing progeny, indicating that these cells are dividing. We have also stained tissues with PH3 and have included an image of a telophase dlRNAi expressing CB NB at 48 hours APF (Fig. S2F).

1. It is unclear whether Notch signaling directly or indirectly regulates temporal transitions. One possibility is that knockdown of Notch signaling decreases cell-cycle speed leading to delayed temporal transitions. The authors should test whether Notch KD affects cell cycle speed using EdU incorporation or PH3 staining. This could be done best using Notch mutant MARCM clones as wt NBs can be used as controls.

We have quantified the number of PH3 positive CB NBs during wandering L3 stages in control and dlRNAi animals. We find that dlRNAi CB NBs are indeed proliferating at reduced rates compared to controls. To test whether reduced cell cycle times are causative for termination delay, we expressed a constitutively active form of PI3-kinase in dlRNAi animals to drive cell growth and proliferation. We found that CB NBs still ectopically persist (Fig. S2E-G).

We have included the following in the text:

“Defects in timing of temporal transitions could be due to defects in cell cycle progression, although embryonic NBs still transition independent of cell division (Grosskortenhaus et al., 2005). We used PH3 to assay CB NB mitotic activity. In Delta knock down animals, the percentage of PH3 positive CB NBs was reduced compared to control (Fig. S2E). At 48 h APF however, Delta knock down CB NBs were still dividing based on PH3 expression (Fig. S2F). To determine whether CB NBs ectopically persist due to defects in cell cycle rate, we co-expressed dp110 to constitutively activate PI3-kinase in Delta knock down animals. A significant number of pcnaGFP expressing, Dpn positive CB NBs were still observed, suggesting that defects in cell cycle timing and growth rates alone cannot account for ectopic persistence of CB NBs into later developmental stages and adulthood (Fig. S2G).”

1. Cas is expressed in NBs either during quiescence and shortly after quiescence. It is possible that the maintenance of Cas in Figure 5D, E is due to NBs that have not re-entered the cell-cycle or have exited quiescence with a strong delay.

Knockdown of Notch pathway has no effect on CB NB reactivation from developmental quiescence. In fact, low levels of Notch are required for CB NBs to reactivate in response to dietary nutrients (Sood, 2022).

Indeed, the authors have previously shown that Notch signaling is important for NB cell cycle reentry during early larval stages (PMID: 35112131). Are Cas and Svp also maintained in late larval N-/MARCM clones (MARCM clonew are made after quiescence exit)?

We have not assayed Cas or Svp expression past 48 hours ALH.

1. The authors have revisited some previously published RNA-seq data showing that Delta is temporally regulated in NB lineages. This is not clearly shown by the authors that the same is true at the protein level.Moreover, they find that mis-expression of late temporal factors or Imp knockdown in early larval brains appear to decrease Delta expression. Such semi-quantitative analysis of gene expression by immunostainings in different conditions can be a bit complicated and not very convincing because variations on intensity levels can be due to slight variations in antibody concentration, or different parameters of image acquisition.

We totally agree, but in this case the difference compared to controls was so readily apparent, that we felt it was not necessary to carry out experiments in clones. All images were acquired with the same confocal settings, experiments were repeated, and we consistently observed the same results. The data shown in Fig. 7D-G is representative.

I suggest that the authors use clonal analysis rather than pan-neuroblast manipulation in order to have internal controls. For example, blocking temporal progression in Syp-RNAi clones (MARCM or Flp-out) and/or svp MARCM clones should lead to maintenance of Imp expression in late larval clones and maintenance of high levels of Delta, which would be easily assessed compared to surrounding NBs.Minor points:Fig 5: the sequential expression of Cas and Svp expression in larval NBs was first described by Maurange et al. 2008. Please cite appropriately.

We have now added the requested citation to the following:

“Over time, the percentage of Cas expressing CB NBs declined, while Svp expressing CB NBs modestly increased (Fig. 5B). Less than 1% of CB NBs co-expressed Cas and Svp at any stage and expression of both factors was absent by 48 hours ALH (Fig. 5B,C). This is consistent with work published previously (Isshiki et al., 2001; Tsuji et al., 2008; Chai et al., 2013; Maurange et al., 2008; Ren et al., 2017; Syed et al., 2017).”

Fig 6A: Please indicate which immunostainings are shown in the overlay panels.

Good catch! We have modified the figure.

P9: "Delta co-immunoprecipitated with Imp.": Add "Delta mRNA co-immunoprecipitated with Imp in RIP-seq experiments" Otherwise, it suggests that you are talking about the protein.

Done

The scheme in Figure 7H is rather complicated to understand. In my opinion, it does not clearly convey the idea that Notch signaling favors the Imp-to-Syp transition.

We have made a new model figure.